



**Temperature variability in southern Europe over the past 16,500 years**

**constrained by speleothem fluid inclusion water isotopes**

**Juan Luis Bernal-Wormull[1], Ana Moreno[1], Yuri Dublyansky[2], Christoph Spötl[2],**

**Reyes Giménez[1], Carlos Pérez-Mejías[3], Miguel Bartolomé[4,5,6], Martin**

**Arriolabengoa[7], Eneko Iriarte[8], Isabel Cacho[9], Richard Lawrence Edwards[10], and**

**Hai Cheng[3,11,12]**

*[1] Department of Geoenvironmental Processes and Global Change, Pyrenean Institute of Ecology*

*(IPE-CSIC), Avda. Montañana 1005, 50059 Zaragoza, Spain. jbernawo7@gmail.com;*

*amoreno@ipe.csic.es; reiesgimenez@gmail.com*

*[2]Institute of Geology, University of Innsbruck, 6020 Innsbruck, Austria.*

*yuri.dublyansky@uibk.ac.at; christoph.spoetl@uibk.ac.at*

*[3] Institute of Global Environmental Change, Xi'an Jiaotong University, Xi'an, 710049, China.*

*perezmegias@xjtu.edu.cn; cheng021@xjtu.edu.cn*

*[4] Departamento de Geología, Museo Nacional de Ciencias Naturales (CSIC), C. de José*

*Gutiérrez Abascal, 2, 28006 Madrid, Spain. mbartucar@gmail.com*

*[5] Swiss Institute for Speleology and Karst Studies (SISKA), Rue de la Serre 68, 2300 La Chaux-*

*de-Fonds, Switzerland.*

*[6] Department of Earth Sciences, Geological Institute, NO G59, Sonneggstrasse 5, ETH, 8092*

*Zurich, Switzerland.*

*[7] Department of Geology, University of the Basque Country, Leioa, Spain.*

*martin.arriolabengoa@ehu.eus*

*[8] Laboratory of Human Evolution-IsoTOPIK Stable Isotope Laboratory, Department of History,*

*Geography & Communication, Edificio de I+D+i, Universidad de Burgos, Pl. Misael Bañuelos*

*s/n, 09001, Burgos, Spain. eiriarte@ubu.es*

*[9] GRC Geociències Marines, Universitat de Barcelona, 28080 Barcelona, Spain. icacho@ub.edu*

*[10] Department of Earth and Environmental Sciences, University of Minnesota, Minneapolis, MN*

*55455, USA. edwar001@umn.edu*

*[11] State Key Laboratory of Loess and Quaternary Geology, Institute of Earth Environment,*

*Chinese Academy of Sciences, Xi'an, 710061, China.*

*[12] Key Laboratory of Karst Dynamics, MLR, Institute of Karst Geology, CAGS, Guilin, 541004,*

*China.*

**Correspondence:** Ana Moreno (*amoreno@ipe.csic.es*)





**ABSTRACT**

In the Northern Hemisphere, the last 16.5 kyr were characterized by abrupt temperature transitions during stadials, interstadials, and the onset of the Holocene. These changes are closely linked to large-scale variations in the extent of continental ice-sheets, greenhouse gas concentrations, and ocean circulation. The regional impact of these rapid climate changes on Southwestern European environments is recorded by various temperature proxies, such as pollen and chironomids preserved in lake sediments. Speleothems and their fluid inclusions serve as valuable proxies, offering high-resolution chronologies and quantitative records of past temperature changes. These non-biogenic quantitative temperature records are essential to assess whether climate models can accurately simulate regionally divergent climatic trends and for understanding global and regional climate mechanisms in the past. Here, we present a record from five speleothems from two caves on the northeastern Iberian Peninsula (Ostolo and Medukilo caves). Using hydrogen isotopic composition of fluid inclusions, we developed a $\delta^2$H/T transfer function in order to reconstruct regional temperatures over the past 16.5 kyr (Ostolo-Mendukilo Fluid Inclusion Temperature record [OM-FIT]). Our findings reveal an increase of 6.0 ± 1.9 ºC at the onset of Greenland Interstadial 1, relative to the cold conditions of the preceding Greenland Stadial 2.1a. Also, the OM-FIT record shows a temperature decline of approximately 5.3 ± 1.9 ºC during the early phase of Greenland Stadial 1. The end of this cold phase and the onset of the Holocene are marked by a rapid warming of about 3-4 °C and reaching a maximum at 11.66 ± 0.03 kyr BP. The OM-FIT record also exhibits abrupt events during the last deglaciation and the Holocene, which are also reflected in the $\delta^{18}$O values of the calcite, including Heinrich Event 1, Greenland Interstadial 1d, and the 8.2 kyr event.

## 1. INTRODUCTION

The last deglaciation in the Northern Hemisphere (ca. 16.5 - 11.7 thousand years [kyr] BP - before present; present = 1950) was punctuated by a series of abrupt climatic changes driven by variations in the extent of large continental ice sheets, greenhouse gas concentrations, and deep-water ocean circulation (Clark et al., 2012). The Holocene was also characterized by variability in terms of temperature, precipitation seasonality, and glacier extent (Wanner et al., 2008, 2011), albeit at much smaller amplitudes compared to the Late Pleistocene. Reconstructing such paleoclimate changes quantitatively poses





significant challenges due to the scarcity of quantitative techniques and the fact that proxy

signals in archives may be influenced by more than one meteorological variable (e.g.,

temperature and precipitation), which complicates our understanding of past temperature

variations (Heiri et al., 2014a; Moreno et al., 2014). These limitations greatly hinder the

assessment of whether reconstructed paleotemperatures in different regions are reflecting

climate variations or different methodologies. Therefore, it is crucial to obtain proxy data

that accurately reflect quantitative changes in paleotemperature, independent of past

changes in rainfall or humidity. Quantitative temperature reconstructions are needed to

assess the ability of climate simulation models to predict regionally divergent trends in

climate change and to better understand the mechanisms of global and regional climate

variability (e.g., Affolter et al., 2019).

The last deglaciation in the Northern Hemisphere involved major climatic shifts

associated with Greenland Stadials (GS-2.1a and GS-1) and Interstadials (GI-1 and the

onset of the Holocene). The impact of these rapid climate changes on Southwestern

European environments is recorded by temperature proxies, e.g., pollen, speleothems,

planktonic foraminifera, and chironomids (Millet et al., 2012; Heiri et al., 2014b;

González-Sampériz et al., 2017; Tarrats et al., 2018; Català et al., 2019; Cheng et al.,

2020). However, the available temperature reconstructions exhibit large regional climate

differences across Europe (Renssen and Isarin, 2001; Heiri et al., 2014b; Affolter et al.,

2019). For example, the chironomid study by Heiri et al. (2014b) revealed that

temperature variations during the last deglaciation were more pronounced in Western

Europe than in Southwestern, Central, and Southeastern Europe. Similar regional

disparities are observed during the Holocene, where the long-term evolution of global and

hemispheric temperature variations remains a subject of debate, with climate models and

proxy records showing differing trends (Marcott et al., 2013; Shakun, 2018; Affolter et

al., 2019). Given these uncertainties, quantitative studies using inorganic archives, such

as fluid inclusions (FI) in speleothems (Dublyansky and Spötl, 2009; Demény et al., 2016,

2021) are gaining increasing relevance (Affolter et al., 2019; Wilcox et al., 2020; Honiat

et al., 2023) as a complement to existing studies largely based on biological archives. The

strengths of this method are: (a) the accurate and precise chronology provided by

speleothems, (b) the well-established link between cave interior temperature and mean

outside air temperature, and (c) the relationship between temperature and water isotopes,

which is controlled by physical rather than biological processes. FI water isotopes can be



measured using different analytical techniques (Vonhof et al., 2006; Dublyansky and Spötl, 2009; Arienzo et al., 2013; Affolter et al., 2014) and FI-based paleotemperature reconstruction methods (Demény et al., 2016, 2021; Uemura et al., 2020). One such approach is based on the $\delta^2H_{FI}$ composition and uses the $\delta^2H_{FI}$-temperature relationship determined for a given study area (Affolter et al., 2019). The principal advantage of this method lies in its reliance on a relatively simple and robust analytical method. The $\delta^2H_{FI}$-temperature relationship is established using monitoring data, and the approach is most effective in settings where $\delta^2H$ variability in rainfall is driven by surface temperature (Demény et al., 2021). For the FI water isotope thermometry method to yield reliable results, four aspects must be considered: (i) FIs must be of primary origin, well-sealed, and sufficiently abundant; (ii) the choice of the transfer function converting the hydrogen and/or oxygen isotope signal ($\delta^2H_{FI}$, $\delta^{18}O_{FI}$) into temperature may bias temperature estimates; (iii) the relationship between $\delta^2H_{FI}$ and $\delta^{18}O_{FI}$ may have changed over time; and (iv) the FI water isotope method assumes that speleothem calcite was deposited under isotopic equilibrium conditions.

Here, we assess the air temperature evolution in the northern Iberian Peninsula over the last 16.5 kyr using quantitative FI-based data from five well-dated stalagmites that overlap during the last deglaciation and Holocene, showing very similar stable isotope trends. This record (dubbed OM-FIT) in conjunction with other regional terrestrial proxy records allows to better disentangle the effects of temperature and humidity reported by previous studies using calcite stable isotope data from caves in southwestern Europe (Bernal-Wormull et al., 2021, 2023). The paleotemperature data obtained from FIs in speleothems represent the first quantitative air temperature reconstruction for northeastern Iberia during the last deglaciation and provide a basis for future studies aiming to enhance our quantitative understanding of rapid regional climate changes.

## 2. STUDY SITES

### 2.1. Ostolo and Mendukilo caves

Ostolo (43°11'16"N, 1°43'56"W, 248 m a.s.l.) and Mendukilo (42°58'25"N, 1°53'45"W, 750 m a.s.l.) caves are located in northern Iberia (Fig. 1). Although only about 28 km apart, they exhibit different geological, geomorphological, and climatic settings. Ostolo cave is located in the Bidasoa river valley, formed within the Carboniferous



133 limestones of the Cinco Villas Massif (Basque Mountains, Western Pyrenees). Mendukilo

134 cave, on the other hand, is developed in Lower Cretaceous limestones (Urgonian, Albian-

135 Aptian) along the eastern boundary of the Basque-Cantabrian basin. For additional details

136 on the caves and the locations of the sampled stalagmites, see Bernal-Wormull et al.

137 (2021, 2023).

138 The climate in the study region is dominated by the Atlantic Ocean, characterized by

139 temperate summers, evenly distributed rainfall throughout the year, and no distinct dry

140 season (Cfb of the Köpper-Geiger climate classification). Mediterranean fronts may also

141 be secondarily responsible for rainfall. Mean annual air temperature (MAAT) and mean

142 annual precipitation are higher in the Ostolo cave area (13.5 ± 0.8 °C; >2000 mm/year)

143 compared to Mendukilo (12.2 ± 0.4 °C; ~1365 mm/year). This temperature difference is

144 even more pronounced inside the caves: the average annual cave air temperature in Ostolo

145 is 13 ºC, while in Mendukilo, it is 8.8 ºC. The lower temperature inside Mendukilo is due

146 to its more closed and hence less ventilated nature compared to Ostolo, which also

147 contains a cave stream that helps stabilize its internal temperature (Bernal-Wormull et al.,

148 2021). In contrast, the "cold-trap" behavior of Mendukilo is consistent with its more

149 complex geometry, resulting in an anomalously low temperature (Bernal-Wormull et al.,

150 2023). The vegetation around both caves is dominated by oak (*Quercus robur* and

151 *Quercus pyrenaica*), alder (*Alnus glutinosa*), beeche (*Fagus sylvatica*), as well as

152 Atlantic-type polycultures, ferns, and heathers.

154 **2.2. Isotopic composition of drip waters in Ostolo and Mendukilo**

155 Quantitative reconstruction of past climate variability from speleothem isotope

156 records relies on understanding the modern vadose karst flow regime (Lachniet, 2009).

157 For Mendukilo, the $\delta^{18}O$ and $\delta^2H$ values of drip waters feeding the stalagmites studied

158 here remain relatively constant, with mean values of −7.7 ± 0.4‰ and −45.3 ± 2.9‰

159 Vienna Standard Mean Ocean Water (VSMOW), respectively (1σ uncertainty), and lack

160 of a seasonal pattern (Bernal-Wormull et al., 2023). The monitoring period in Mendukilo

161 cave lasted nearly three years, with measurements taken every 2-3 months (2018-2021).

162 In Ostolo, the $\delta^{18}O$ and $\delta^2H$ values of drip water are also similarly stable, with mean

163 values of −6.3 ± 0.2‰ and −37.8 ± 1.6‰ VSMOW, respectively, with carbonate

164 precipitation throughout the year in only one gallery of the cave (Bernal-Wormull et al.,

165 2021). The monitoring interval in Ostolo was 3-4 months over one year (2019-2020).



### 2.3. Isotopic composition of rainfall

The rainfall stable isotopic composition near the study sites was analyzed by Giménez et al. (2021) on an event basis above "Las Güixas" cave (Villanúa village), approximately 100 km east of the Ostolo and Mendukilo caves. This show cave, located in the Central South Pyrenees (Fig. 1), experiences a transitional Mediterranean-Oceanic climate (Cfb of the Köpper-Geiger climate classification) with a MAAT of 11 ºC and around 1100 mm of annual precipitation. During the winter the westerly winds and Atlantic fronts are responsible for most rainfall, while rest of the year is mixed between Mediterranean and Atlantic fronts (Giménez et al., 2021), similar to the conditions in the area of Ostolo and Mendukilo caves. Two years of stable isotope data in precipitation and air temperature on an event scale are available from this station (2017-2019, Giménez et al., 2021). The weighted mean values of $\delta^{18}$O and $\delta^{2}$H are $-7.8 \pm 4.3$‰ and $-54.5 \pm 32.9$‰, respectively, with seasonal variations reaching total amplitudes of 23 and 174‰, respectively (Giménez et al., 2021). The Local Meteoric Water Line (LMWL) is defined as $\delta^{2}$H = $7.56 \cdot \delta^{18}$O + 4.33 (n = 210; $R^2$ = 0.97). The slope of the LMWL is close to that of the Global Meteoric Water Line (GMWL; Rozanski et al., 1993) and aligns well with the water line defined by the drip waters of Mendukilo and Ostolo caves (Fig. 2A). In general, the isotopic composition of rainfall correlates with air temperature for the 2-year period (n =210; $R^2$ = 0.44, Fig. 2B), and show moderate correlation with relative humidity and a weaker correlation with rainfall amount at event scale when performing a Spearman's correlation ($r_s$; n = 180; between rainfall amount and $\delta^{18}$O [$\delta^{2}$H]: $r_s$ = $-0.27$ [$-0.25$]; between temperature and $\delta^{18}$O [$\delta^{2}$H]: $r_s$ = 0.70 [0.69]; between relative humidity at the rainfall site and $\delta^{18}$O [$\delta^{2}$H]: $r_s$ = $-0.46$ [$-0.41$]) (Giménez et al., 2021).

### 3. METHODS

### 3.1 Sampling and petrography

Stalagmites OST1, OST2 and OST3 were retrieved from a gallery in Ostolo cave, where active speleothem deposition was not observed. Stalagmites MEN-2 and MEN-5 were retrieved from a gallery in Mendukilo cave, where active calcite precipitation was only observed at the original dripping point of MEN-5. See Bernal-Wormull et al. (2021, 2023) for more details on these caves. All stalagmites were cut longitudinally and the





central slab was polished. Small blocks were cut along the growth axis for the preparation

of doubly-polished thin sections (about 200 μm). FIs were studied in these thin sections

using a Nikon Eclipse transmitted-light microscope.

**3.2. FI stable isotopic composition**

A total of 344 carbonate subsamples (including duplicates) were crushed and

analyzed for $\delta^2H_{FI}$ (287 subsamples of Mendukilo stalagmites and 69 of Ostolo samples).

Between 0.3 and 2.5 g of calcite were used to ensure a sufficiently high water yield (0.1-

1 μL). Stable isotope measurements were performed using a Delta V Advantage isotope

ratio mass spectrometer, following the methodology described by Dublyansky and Spötl,

(2009). $\delta^2H_{FI}$ values are reported in per mil relative to VSMOW. The average long-term

precision of replicate measurements of an in-house calcite standard is ±2.7 ‰ for $\delta^2H_{FI}$

for water amounts between 0.1 and 1 μL.

$\delta^2H_{FI}$ is regarded as a more robust proxy of paleotemperature than $\delta^{18}O_{FI}$, as it is

less influenced by non-climatic parameters, with no other sources of hydrogen affecting

the water trapped in the calcite (Demény et al., 2016, 2021; Affolter et al., 2019). In

addition, $\delta^{18}O_{FI}$ values obtained with the Innsbruck FI setup can be inaccurate for samples

of low water content (<0.1 μL; Dublyansky and Spötl, 2009 ). Therefore, we only used

$\delta^2H_{FI}$ values in this study.

**4. RESULTS**

**4.1. Petrography**

The Ostolo and Mendukilo stalagmites consist of coarse crystalline calcite and are

macroscopically homogenous without any sign of recrystallization. The MEN-2 and

MEN-5 stalagmites exhibit a columnar fabric, lack growth hiatuses, and do not show

macroscopically visible laminae (Bernal-Wormull et al., 2023). In contrast, the Ostolo

stalagmites shows a more porous columnar microcrystalline fabric that transitions into an

elongated-columnar type (Bernal-Wormull et al., 2021). Two hiatuses are present in

OST3, marked by organic inclusions and micrite layers.

Primary FIs were observed in all stalagmites samples (Fig. 3). The Mendukilo

samples contain considerably more FIs compared to those from Ostolo, mainly

concentrated along growth layers (Fig. 3A). In the Mendukilo stalagmites, primary inter-





crystalline (10-30 μm; Fig. 3B) and intra-crystalline (10 to >100 μm; Fig. 3C) FIs are

discernible. These intra-crystalline primary FIs are elongated and rounded or pyriform in

shape (rounded at the base with a spike extending in the speleothem growth direction;

Fig. 3C; Lopez-Elorza et al., 2021). In Ostolo, FIs are less prominent and are mostly intra-

crystalline, located along or around white porous laminae and within the more elongated

columnar or microcrystalline fabrics (Fig. 3D, E). The intra-crystalline FIs in Ostolo

samples are, on average, smaller than those in the Mendukilo stalagmites (10-40 μm) and

predominantly exhibit pyriform or rounded shapes (Fig. 3F). Petrographic observations

confirm that the FIs in these samples are primary, well preserved, and suitable for their

stable isotopic analysis.

**4.2. Last deglaciation and Holocene δ$^{18}$O speleothem record**

The chronology of the Ostolo stalagmites spans the last deglaciation between 16.5

and 11.7 kyr BP with high precision due to their very high $^{238}$U concentrations (10-80

246 ppm). The carbonate δ$^{18}$O (δ$^{18}$O$_c$) profiles show consistency among the three stalagmites

(Fig. 4). OST1 and OST2 have more negative values (−5 to −8.9‰) during GS-1 and GS-

2.1a, and less negative values (up to −3.4‰) during GI-1 and the onset of the Holocene.

OST3 did not grow during the intervals characterized by the most negative δ$^{18}$O$_c$ values

recorded by the other two stalagmites (Fig. 4). On the other hand, the MEN stalagmites,

despite having lower $^{238}$U concentrations (100-350 ppb), also have lower detrital $^{232}$Th

contents, enabling robust age models for both stalagmites. These models cover various

intervals of the Holocene and GS-1 with good overlap (Fig. 4), specifically: (i) MEN-2

grew between 12.8 and 6.3 kyr BP, with δ$^{18}$O values that remain stable during GS-1,

followed by an abrupt increase, reaching the highest values of the entire record at the GS-

1/Holocene transition (from −5.2‰ in GS-1 to −4.3‰ at 11.6 kyr BP). (ii) MEN-5 spans

the last 8.8 kyr and presents prominent negative values during certain short events (e.g.,

8.2 kyr BP with a value of −6.3‰, replicated by MEN-2), which are synchronous, within

age uncertainties, with abrupt changes in the isotopic composition of North Atlantic

surface waters (Kleiven et al., 2008; Carlson et al., 2008). More details on the chronology

and isotopic data of these speleothems are provided by Bernal-Wormull et al. (2021,

2023).

**4.3. FI isotopes**



OST samples are characterized by variable water content, with replicates yielding
a mean standard deviation of ±2.7‰ for $\delta^2H$. We assigned this value to individual
measurements as an overall uncertainty estimate. Not all OST samples could be
duplicated due to sometimes low water amounts and petrographically complex FI
assembles in some samples (Fig. 3D, E), which restricted subsampling of some individual
growth layers. All MEN measurements were duplicated, triplicated, or even
quadruplicated. The $\delta^2H$ values of sub-samples of MEN-2 and MEN-5 (ranging between
272     −34 and −61‰) with water contents of 0.1 to 1 μL replicated within 2.7‰.

$\delta^2H_{FI}$ values for the Holocene and GI-1 are comparable to cave drip waters at
Mendukilo and Ostolo caves (Fig. 4). In contrast, values are more negative during GS-1
and GS-2.1a (Fig. 4). GS-2.1a is represented by 8 OST subsamples with a mean $\delta^2H_{FI}$
value of −58‰. One of these values, dated to 15.80 ± 0.05 kyr BP, is even more depleted
(−66.8 ± 2.4‰). Values become less negative rapidly at 14.57 ± 0.05 kyr BP (Fig. 4;
mean during GI-1: −40‰). This trend is interrupted in the three OST stalagmites at 14.13
± 0.09 kyr BP, leading to more negative values (between −40 and −56‰). During GS-1,
the $\delta^2H_{FI}$ values decrease again (Fig. 4), averaging −51‰ before showing a rapid increase
at the onset of the Holocene (−36‰). The MEN-2 record also shows a mean of −51‰
during GS-1, though the transition to the Holocene is more gradual. Between 8.7 and 6.3
283     kyr BP, MEN-2 and MEN-5 $\delta^2H_{FI}$ values show excellent correlation (Fig. 4). There is no
significant variation between the Greenlandian (−44‰), Northgrippian (−43‰), and
Meghalayan (−42‰). Despite these relatively stable $\delta^2H_{FI}$ values throughout the
Holocene substages, a short negative shift is identified at 8.29 ± 0.07 (−54.9 ± 6.5‰) kyr
BP.

**5.  DISCUSSION**

**5.1. Interpretation of the $\delta^{18}O$ signal**

Variations in stalagmite $\delta^{18}O_c$ records may reflect changes in the $\delta^{18}O$ of surface
ocean waters from the moisture source area as well as changes in atmospheric processes
which control the fractionation of oxygen isotopes in route to the site where rainfall
occurs (McDermott, 2004; Lachniet, 2009). In the Ostolo stalagmites, the $\delta^{18}O_c$ signal is
coherent with air temperature changes throughout the deglaciation period (Bernal-
Wormull et al., 2021). The overall $\delta^{18}O_c$ pattern observed in these stalagmites is similar



to that of speleothems from the Pyrenees (Bartolomé et al., 2015; Cheng et al., 2020) and

the Alps (Luetscher et al., 2015; Li et al., 2020), which also predominantly receive

Atlantic-derived moisture and where $\delta^{18}O_c$ primarily reflects atmospheric temperature.

Superimposed on the temperature effect are changes in the isotopic composition of

seawater, which may account for the negative excursion in the Ostolo $\delta^{18}O_c$ record during

Heinrich event 1 (HE1) at 16.2–16.0 kyr BP, with values reaching as low as −8.9‰

(Bernal-Wormull et al., 2021; Fig. 4).

Conversely, the MEN $\delta^{18}O_c$ record captures a temperature signal that is obscured

by the influence of rainfall amount, since temperature and humidity changes may have

competing effects on the $\delta^{18}O_c$ signal (Bernal-Wormull et al., 2023). Additionally, during

the earlier part of the record (13-8 kyr BP), changes in the oceanic isotopic composition

associated with meltwater input (Skinner and Shackleton, 2006; Eynaud et al., 2012) that

further affect the signal. A prominent feature of the MEN-2 and MEN-5 $\delta^{18}O_c$ records is

a −0.7‰ anomaly (relative to the Holocene mean of −5.4‰) observed at 8.11 and 7.00

312    kyr BP (Fig. 4). These two events of anomalously low $\delta^{18}O_c$ values likely reflect rapid,

short-lived decreases in temperature and in the $\delta^{18}O$ of the surface ocean water, rather

than increased rainfall, as proposed in previous studies (e.g., LeGrande and Schmidt,

2008; Domínguez-Villar et al., 2009; Matero et al., 2017; García-Escárzaga et al., 2022).

**5.2. Isotope-temperature conversion**

The composite paleotemperature records of the Ostolo and Mendukilo

speleothems are based on 356 FI samples (and replicates), applying a regional water

isotope-temperature relationship derived from monitoring data (isotopic data of drip

water and outside temperature) of both caves (Bernal-Wormull et al., 2021, 2023) and the

relationship between rainfall $\delta^2H$ ($\delta^2H_r$) and modern air temperature. The latter provides

a relationship between air temperature and the stable isotopic composition of rain ($\delta^{18}O_r$

and $\delta^2H_r$) observed from July 2017 to June 2019 (n = 210). The observed correlation

between $\delta^{18}O$ and air temperature is verified at biannual scale, with significant correlation

between MAAT and the weighted average $\delta^{18}O_r$, based on a multiple regression model

using a univariate Spearman's correlation between $\delta^{18}O_r$ and air temperature at the time

of precipitation (same data series), that also accounts for rainfall amount and relative

humidity ($r_s = 0.7$; $p \ll 0.01$, Giménez et al., 2021).



$\delta^{18}O$ and $\delta^2H$ values of seawater vary on glacial-interglacial timescales due to the

ice-volume effect: When surface waters evaporates from the ocean, lighter stable isotopes

are preferentially removed into the vapor phase, leading to increased $\delta^{18}O$ and $\delta^2H$ values

in the ocean water as more fresh water is stored as ice on continents (Lachniet, 2009).

$\delta^2H_{FI}$ values were corrected for the ice-volume effect during the deglaciation period

covered by the MEN and OST speleothems. This correction used a gradient derived for

$\delta^{18}O$ (Bintanja et al., 2005) converted to $\delta^2H$ using a factor of eight. Paleotemperatures

were then estimated using a linear $\delta^2H/T$ transfer function anchored to the MAAT at both

cave sites and the isotopic composition of drip water ($\delta^2H_d$; Ostolo $\delta^2H_d = -37.8‰$;

Mendukilo $\delta^2H_d = -45.3‰$), with corrections for the elevation of the Villanúa monitoring

station (950 m a.s.l.). The modern $\delta^2H$ values were adjusted for the elevation difference

between the rainfall sampling station and the studied caves, assuming a lapse rate of 0.2‰

342    per 100 m for $\delta^{18}O$, i.e., 1.6‰ per 100 m for $\delta^2H_p$ (Poage, 2001). The uncertainties

associated with $\delta^2H_{FI}$, $\delta^2H_d$, $\delta^2H/T$, and MAAT, as well as the slope of the LMWL, were

propagated through the calculation steps. Due to a lack of constraints on past seasonal

changes in precipitation and effective infiltration, we assume constant annual infiltration

over time.

**5.3.OM-FIT: paleothermometric record derived from FI stable isotope data**

Our $\delta^2H_{FI}$ values provides a robust record, because: (i) part of the record is well

replicated by samples from two caves from different climatic settings (e.g., during the

Younger Dryas [YD]), (ii) stalagmites from the same cave are replicated (within their

respective uncertainties), and (iii) a large proportion of the samples have multiple

replications. We investigated the temperature dependence of the hydrogen (and oxygen)

isotope composition of precipitation water in the study region, examining the modern-

355    day $\delta^2H/T$ and $\delta^{18}O/T$ gradients. This relationship, which may change over time, was

examined by Rozanski et al. (1992) for Central Europe and applied by Affolter et al.

(2019) to a 14 kyr record from Milandre cave (Switzerland). It was similarly applied to

Last Interglacial records from Alpine caves (Wilcox et al., 2020; Honiat et al., 2023). The

relationship between mean annual $\delta^{18}O_r$ and MAAT ($\delta^{18}O/T$) is $0.55 \pm 0.03‰$ °C$^{-1}$ for

the "Las Güixas" tourist cave in Villanúa, which is consistent with the average European

$\delta^{18}O/T$ gradient of $0.59 \pm 0.08‰$ °C$^{-1}$ (Rozanski et al., 1992). The OST and MEN FI

isotope data overlap chronologically for the YD, allowing for their combination into a



single temperature transfer function (OM-FIT) covering the last 16.7 kyr BP (Fig. 5). The

OM-FIT is calculated using the corrected $\delta^2H_{FI}$ values, $\delta^2H_d$, MAAT ($T_{modern}$), and the

modern-day $\delta^2H/T$ gradient derived from the LMWL of rainfall isotopes:

$$T_{OM-FIT} = T_{modern} - \frac{\delta^2H_d - \delta^2H_{FI\,(corrected)}}{\delta^2H/T_{gradient}} \tag{1}$$

As explained above in chapter 5.2, the $\delta^{18}O$ values were further adjusted using the

equilibrium fractionation factor of eight to elaborate the temperature reconstruction

exclusively with $\delta^2H$ data. The temperature reconstruction with Equation (1) is based on

the mean relationship of 4.4‰/°C (for $\delta^2H$). The final calculated uncertainty in the

paleotemperature ranges from 1.8 to 3.0 °C.

**5.4. Temperature regime of Northern Spain based on OM-FIT**

**5.4.1.  Last deglaciation**

The Ostolo cave $\delta^{18}O_c$ and $\delta^2H_{FI}$ records (Fig. 4) and the OM-FIT (Fig. 5) show clear

evidence of rapid temperature changes during GS-2.1a, GI-1, GS-1, and the onset of the

Holocene. The timing and amplitude of these changes are in well agreement with other

European oxygen isotope records from lake sediments (Von Grafenstein et al., 1999; Van

Raden et al., 2013) and speleothems (Luetscher et al., 2015; Affolter et al., 2019; Cheng

et al., 2020; Li et al., 2020). The strong similarity between these records and NGRIP $\delta^{18}O$

(Rasmussen et al., 2014) and temperature reconstructions (Kindler et al., 2014) (Fig. 6)

supports the idea of a common North Atlantic climate forcing during the last deglaciation

on millennial to centennial timescales.

The OM-FIT record suggests that regional MAAT during GS-2.1a was slightly lower

than during GS-1, characterized by a negative excursion at 15.8 ± 0.1 kyr BP and a

temperature decrease of approximately 2.0 °C relative to the GS-2.1a average (Fig. 5).

This OM-FIT anomaly corresponds with the final phase of HE1, related to massive

iceberg discharges from the Laurentide ice sheet, which collapsed around 16.2 ± 0.3 kyr

BP (Landais et al., 2018). Regionally, a significant glacier advance occurred at that time

in the Pyrenees and other Iberian mountains (García-Ruiz et al., 2023), and speleothems

from Meravelles cave (NE Iberia) record a notable $\delta^{18}O_c$ anomaly between 16.2 and 15.9

395    kyr BP (Pérez-Mejías et al., 2021). This anomaly appears to reflect changes in the isotopic





composition of the moisture source, contributing to the negative excursion in the OST2

$\delta^{18}O_c$ record between 16.2 and 16.0 kyr BP (Bernal-Wormull et al., 2021; Fig. 5). This

observation confirms that the OM-FIT record captured not only temperature history on

millennial scales but also abrupt climate events on a centennial scale.

A rapid temperature increase of 6.0 ± 2.1 °C occurred at the onset of GI-1 (Fig. 5).

This increase in the OM-FIT record coincides with an important glacier retreat in the

Iberian mountains (García-Ruiz et al., 2023), an increase in chironomid-inferred July air

temperatures (from ca. 11 °C to ca. 16 °C) from the west-central Pyrenees (Millet et al.,

2012), and an increase in MAAT (from ca. 12.2 °C to ca. 18.6 °C) recorded by branched

glycerol dialkyl glycerol tetraethers in the Padul palaeolake record (Sierra Nevada,

southern Iberian Peninsula; Rodrigo-Gámiz et al., 2022). The onset of GI-1 in the OM-

FIT was recorded by $\delta^2H_{FI}$ data from the OST1 and OST3 stalagmites. The amplitude of

this abrupt warming is in agreement with other European temperature records, such as

estimates based on $\delta^{18}O_c$ data from Alpine speleothems (Luetscher et al., 2015; Li et al.,

2020). Von Grafenstein et al. (2013) used a combination of ostracod, mollusc, and

charophyte data to estimate a rise of about 6 °C in MAAT for this transition at the

Gerzensee lake site. The Ammersee record, using a coefficient derived from a study of

northern Switzerland stalagmites (0.48‰/°C, Affolter et al., 2019), estimated a warming

of about 5.5 °C (4.1–8.4 °C) (Li et al., 2020) for this transition.

During GI-1, the $\delta^2H_{FI}$ record is marked by higher $\delta^2H$ values and similar

temperatures in the OM-FIT record compared to the onset of the Holocene (Fig. 5). As

observed in the OST $\delta^{18}O_c$ record, $\delta^2H_{FI}$ values follow a negative trend towards the end

of GI-1. Within this interstadial, a significant inflection point occurs with a negative

anomaly at 14.1 ± 0.1 kyr BP in the OM-FIT record. This suggests that the OM-FIT

minimum during GI-1, also registered at 14.10 ± 0.03 kyr BP in the OST $\delta^{18}O_c$ record

and equivalent to GI-1d in NGRIP (Rasmussen et al., 2014), involved the most

pronounced cooling of GI-1 (between 3.0 and 3.7 ± 2.1 °C in the OM-FIT record),

occurring just after the GI-1e warm phase (Fig. 5). This cooling event is contemporaneous

with glacier expansions in the Pyrenees (García-Ruiz et al., 2023) and a centennial-scale

cooling at Ech paleolake (Millet et al., 2012), Lake Estanya (Vegas-Vilarrúbia et al.,

2013) and in the Portalet sedimentary sequence (González-Sampériz et al., 2006).

Apparently, this relatively small decrease in temperature during GI-1d, as quantified by

the OM-FIT record and chironomid-inferred July air temperatures (Millet et al., 2012) in





this region, resulted in (i) an important vegetation response (González-Sampériz et al.,

2017), characterized by a decrease in juniper and an expansion of steppe herbs during this

cold and dry event, and (ii) carbonate and massive organic-rich silt deposition during

warm and humid interstadials alternating with siliciclastics under cold and arid conditions

(González-Sampériz et al., 2006).

Between 13.0 and 12.5 kyr B.P., the $\delta^2H_{FI}$ decrease (Fig. 4) records a cooling of 5.5

± 2.1 ℃ in the OM-FIT record (Fig. 5), marking the initial part of GS-1 (Rasmussen et

al. 2014). Similar cooling magnitudes were reported for the central Pyrenees (Bartolomé

et al., 2015). On the other hand, this change appears slightly larger compared to cooling

registered by summer air temperature records of the GI-1/GS-1 transition, such as those

from lake sediments in NW Iberia (2-3 ℃; Muñoz Sobrino et al., 2013) and the central

Pyrenees (1.5-2 ℃; Millet et al., 2012). This important change in the OM-FIT record also

agrees in magnitude with a rapid cooling recorded by (i) speleothems from the Alps

(around 4–5 °C; Li et al., 2020) and the Jura Mountains (4.3 ± 0.8 °C; Affolter et al.,

2019), and (ii) a drop in sea-surface temperatures of around 4 °C off the Iberian coast at

12.9 kyr BP (Rodrigues et al., 2010; Martrat et al., 2014).

The end of the GS-1 cold phase and the onset of the Holocene are marked by a rapid

warming in the OM-FIT record of about ∼4 °C (Fig. 5), peaking at 11.67 ± 0.02 kyr BP.

The variability of MEN $\delta^{18}O_c$ data during the GI-1/GS-1 and GS-1/Holocene onset

transitions is less pronounced compared to OST $\delta^{18}O_c$. This observation may be due to

the proximity of Mendukilo cave to the Atlantic coast, with temperature and humidity

changes having competing effects on $\delta^{18}O_c$, as already reported in other speleothem

records from this region (e.g., Baldini et al., 2019). In contrast, the $\delta^{18}O_c$ of speleothems

from Pyrenean caves is predominantly controlled by temperature (Bartolomé et al., 2015;

Cheng et al., 2020; Bernal-Wormull et al., 2021), resulting in a more "smoothed"

temperature signal compared to the OST $\delta^{18}O_c$ record during GS-1, a cold and dry period

(Fletcher et al., 2010). Nevertheless, the MEN $\delta^2H_{FI}$ records captures important changes

during the GI-1/GS-1 and GS-1/Holocene transitions and correlates quite well with the

$\delta^2H_{FI}$ data from OST (Fig. 4).

**5.4.2. Holocene**

As mentioned above, the Holocene section of the OM-FIT record (Fig. 5) is based on

$\delta^2H_{FI}$ values of the MEN stalagmites (Fig. 4). This record not only captures variability in



$\delta^{18}O_c$ composition influenced by temperature but also reflects past hydroclimatic

conditions (Bernal-Wormull et al., 2023). This observation introduces a limitation in

reconstructing periods of relatively stable temperature, such as the Holocene, which is

represented by centennial-scale OM-FIT temperature variability that reaches up to 2 °C

in certain intervals. However, these variations are close to the uncertainty range of the

OM-FIT record (± 1.8 °C to ± 3.0 °C for the Holocene). Therefore, these reconstructed

quantitative temperature data for the Holocene must be viewed with caution. On

millennial scales, the OM-FIT record shows peak temperatures during the onset of the

Holocene (until ~10 kyr BP), albeit with high variability. This early rapid warming is

also recorded by the hydroclimate-sensitive isotopic signal of the SIR-1 stalagmite from

NW Iberia (Rossi et al., 2018). This observation underscores the value of obtaining a

temperature-sensitive record in regions where the isotopic signal of speleothems is also

influenced by the amount effect, such as the MEN $\delta^{18}O_c$ record.

The OM-FIT record does not capture a clear cooling trend after the Holocene Thermal

Maximum (HTM) compared to the $\delta^{18}O$ record from Greenland ice cores (Rasmussen et

al., 2014) and the Milandre cave fluid inclusion temperature record (MC-FIT) record from

central Europe (Affolter et al., 2019) (Fig. 6), instead suggesting stable temperatures. This

Neoglacial cooling, widespread across the Northern Hemisphere, is well documented

throughout Europe (e.g., Larocque-Tobler et al., 2010; Ilyashuk et al., 2011) and Iberia

(Sancho et al., 2018; Leunda et al., 2019; Català et al., 2019; García-Ruiz et al., 2020).

The absence of this cooling in the OM-FIT record is likely due to masking by large

centennial variability and large temperature uncertainties. The temperature trends in

MEN $\delta^{18}Oc$ and OM-FIT differ from those captured by chironomids in the central

Pyrenees (Tarrats et al., 2018), which indicate a millennial-scale cooling during the

middle Holocene compared to the HTM and the late Holocene (Fig. 6). This observation

highlights the differences between temperature records derived from speleothems (OM-

FIT, without seasonal bias) and chironomids (recording summer air temperature), as in

the case for GS-1.

Despite the limited precision of OM-FIT, it can identify abrupt centennial events,

some of which are also evident in the $\delta^{18}Oc$ values of MEN-2 and MEN-5 (Fig. 6). For

example, one of the lowest OM-FIT temperatures (9.8 °C) occurred at 11.50 ± 0.08 kyr

BP (mean temperature at the onset of the Holocene, 12.3 ± 1.8 °C), corresponding within

age uncertainties to the Preboreal Oscillation (11.4 kyr) recorded in Greenland ice cores



(11.27 ± 0.03 kyr BP, based on the new ice core chronology - Seierstad et al., 2014) and

by MC-FIT in Switzerland (11.37 ± 0.15 kyr BP - Affolter et al., 2019) (Fig. 6). Another

example is the 9.2-kyr event, documented across the Northern Hemisphere (e.g., Masson-

Delmotte et al., 2005; Genty et al., 2006; Rasmussen et al., 2007; Fleitmann et al., 2008)

ans supported by terrestrial (Carrión, 2002; Vegas et al., 2010; Iriarte-Chiapusso, 2016;

Mesa-Fernández et al., 2018; Baldini et al., 2019) and marine records from Spain (Nebout

et al., 2009). This event is captured by a $\delta^2H_{FI}$ value of −51‰ in MEN-2 and an OM-FIT

temperature of 10.4 ± 1.9 ℃ at 9.29 ± 0.08 kyr BP (Fig. 6). However, it is absent from

the $\delta^{18}O_c$ record of MEN-2, and previous research suggests that the climate in northern

Spain was likely considerably warmer and wetter ~9 ka BP (Morellón et al., 2018; Tarrats

et al., 2018; Baldini et al., 2019). This observation supports the assertion of Bernal-

Wormull et al. (2023) that the less variable $\delta^{18}O_c$ signal in Mendukilo cave is influenced

not only by short-lived decreases in $\delta^{18}O_{sw}$ but also by changes in humidity.

Catastrophic meltwater discharge during the '8.2 kyr event' from glacial lake Agassiz

lowered the isotope composition of North Atlantic surface water by 0.4‰ (Kleiven et al.,

2008; Carlson et al., 2008) and led to a wide-spread cooling across the circum-North

Atlantic. The isotopic signal of this meltwater event was transported by the westerlies and

left an imprint in the isotopic composition of precipitation in Iberia (LeGrande and

Schmidt, 2008; Bernal-Wormull et al., 2023). The 8.2-kyr event overlapped a multi-

centennial cool period from 8.29 to 8.10 ± 0.04 kyr BP recorded by MEN $\delta^{18}O_c$,

characterized by an abrupt drop in temperature of about ~2.7 ℃ between 8.31 ± 0.06 and

8.29 ± 0.07 kyr BP in the OM-FIT record (Fig. 6). This cooling within an interglacial

coincided with significant vegetation changes in the Iberian Peninsula (Allen et al., 1996;

Carrión and Van Geel, 1999; González-Sampériz et al., 2006). This could be important

for assessing future climate conditions in this region if changes in large parts of the

climate system (climate tipping elements; Armstrong McKay et al., 2022) intensify

beyond a warming threshold.

The cooling amplitude during the 8.2 kyr event recorded by OM-FIT appears more

pronounced than in other Northern Hemisphere temperature and precipitation records,

with proxy evidence across Europe indicating a cooling by ~ 1-1.7 ℃ during this event

(Davis et al., 2003; Morrill et al., 2013; Baldini et al., 2019). Other terrestrial records in

southwestern Europe offer important insights into the paleoenvironment during this event

(e.g., Fletcher et al., 2013; González-Sampériz et al., 2017; Morellón et al., 2018;



Zielhofer et al., 2019). Some records often present conflicting insights on humidity
conditions due to the exposure of this study region to both Mediterranean and North
Atlantic air masses (Moreno et al., 2017, 2021). However, most of these terrestrial records
capture broader climate shifts, often lacking the resolution to fully constrain the regional
response to the 8.2 kyr event. It is therefore likely that these long-term changes are more
influenced by local summer insolation than by an Atlantic climatic anomaly, as suggested
by Kilhavn et al. (2022). Thus, other stalagmite records from the region (Kilhavn et al.,
2022) and the combination of the carbon isotopic composition ($\delta^{18}O_c$ and $\delta^{13}C_c$) and the
FI record from Mendukilo stalagmites offers a better understanding of the regional
response during this colder-than-average Holocene period, which was characterized by
increased humidity and changes in moisture source composition (Domínguez-Villar et
al., 2009; Kilhavn et al., 2022; Bernal-Wormull et al., 2023).
**6.   CONCLUSIONS**
The Ostolo and Mendukilo speleothems provide a replicated and precisely dated
record of paleotemperature in NE Iberia for the past 16.5 kyr BP. The OM-FIT record
contributes novel, non-biogenic evidence of rapid temperature transitions during the last
deglaciation and the Holocene, including the identification of abrupt events. Our findings
indicate temperatures for GS-2.1a up to 6.0 ± 1.9 °C lower than those for GI-1 and
present-day conditions, and constrain the regional response of HE-1 between 16.2 and
15.8 kyr BP. The sharp rise in temperatures during the GS-2.1a/GI-1 transition was
quantitatively comparable to other records from SW Europe. Temperatures during GI-1
were equivalent to those of the Holocene, with a minimum observed at 14.1 ± 0.1 kyr BP
during GI-1d. The rapid temperature changes at early GS-1 and the onset of the Holocene
recorded by OM-FIT are consistent with to those reported from other parts of Europe.
Neither $\delta^{18}O_c$ nor OM-FIT reveal significant millennial-scale changes during the
Holocene. The 8.2 kyr event is recorded between 8.29 and 8.10 ± 0.04 kyr in the $\delta^{18}O_c$
record, centered at 8.29 ± 0.07 kyr in the OM-FIT record, synchronous with Greenland
ice-core data and well-dated records from central and southwestern Europe.



**Appendix A**

**Table A1.** FI $\delta^2H$ measurements of Ostolo samples. The $\delta^2H$ values were corrected for the ice-volume effect during the deglaciation period covered by the Ostolo speleothems. Each time span of each sample represents the duration covered by the respective calcite blocks sampled from the stalagmites used for the fluid inclusion measurements (without taking into account the age model uncertainty).

| FI sample | Water amount (µL) | Water content (µL/g) | $\delta^2H$ (‰ VSMOW) measured | Mean $\delta^2H$ (‰ VSMOW) | $\delta^2H$ Std Dev | $\delta^2H$ Error | Mean $\delta^2H$ adjusted for IV (‰ VSMOW) | Age (kyr BP) |
|---|---|---|---|---|---|---|---|---|
| OST1-16.1A | 0.52 | 0.27 | -50.85 | -49.74 | 1.57 | 2.70 | -57.08 | 16.06 ± 0.06 |
| OST1-16.1B | 0.69 | 0.45 | -48.62 | | | | | |
| OST1-15.2A | 0.04 | 0.06 | -54.99 | | | | | |
| OST1-15.2B | 0.18 | 0.18 | -51.05 | -51.43 | 3.38 | 3.38 | -57.98 | 15.16 ± 0.05 |
| OST1-15.2C | 0.87 | 0.31 | -48.26 | | | | | |
| OST1-14.6A | 0.39 | 0.39 | -25.99 | -25.37 | 0.88 | 2.70 | -31.36 | 14.57 ± 0.05 |
| OST1-14.6B | 0.86 | 0.79 | -24.75 | | | | | |
| OST1-14.2A | 0.11 | 0.44 | -43.68 | -43.39 | 0.41 | 2.70 | -49.09 | 14.20 ± 0.02 |
| OST1-14.2B | 0.10 | 0.29 | -43.10 | | | | | |
| OST1-13.0 | 0.57 | 0.37 | -32.51 | -32.51 | n/a | 2.70 | -36.96 | 13.02 ± 0.04 |
| OST1-10.9A | 0.19 | 0.17 | -29.28 | -26.58 | 3.82 | 3.82 | -28.86 | 10.95 ± 0.20 |
| OST1-10.9B | 0.12 | 0.40 | -23.87 | | | | | |
| OST3-16.4 | 0.11 | 0.37 | -46.07 | -46.07 | n/a | 2.70 | -53.59 | 16.40 ± 0.11 |
| OST3-14.3 | 0.21 | 0.18 | -26.03 | -26.03 | n/a | 2.70 | -31.83 | 14.30 ± 0.09 |
| OST3-14.1A | 0.09 | 0.07 | -36.69 | | | | | |
| OST3-14.1B | 0.19 | 0.13 | -33.52 | -35.29 | 1.62 | 2.70 | -40.89 | 14.11 ± 0.09 |
| OST3-14.1C | 0.20 | 0.19 | -35.66 | | | | | |
| OST3-13.5A | 0.14 | 0.13 | -34.48 | | | | | |
| OST3-13.5B | 0.18 | 0.14 | -33.01 | -34.91 | 2.15 | 2.70 | -39.90 | 13.50 ± 0.09 |
| OST3-13.5C | 0.16 | 0.11 | -37.25 | | | | | |
| OST3-13.0 | 0.20 | 0.14 | -30.97 | -30.97 | n/a | 2.70 | -35.43 | 13.00 ± 0.08 |
| OST3-12.9 | 0.12 | 0.12 | -42.50 | -42.50 | n/a | 2.70 | -46.85 | 12.90 ± 0.06 |
| OST3-12.8 | 0.17 | 0.14 | -43.75 | -43.75 | n/a | 2.70 | -47.98 | 12.80 ± 0.08 |
| OST3-11.7A | 0.11 | 0.12 | -26.15 | -24.99 | 1.64 | 2.70 | -27.92 | 11.67 ± 0.02 |
| OST3-11.7B | 0.24 | 0.27 | -23.83 | | | | | |
| OST3-11.6A | 0.09 | 0.08 | -35.16 | -39.68 | 6.39 | 6.39 | -42.60 | 11.60 ± 0.02 |
| OST3-11.6B | 0.14 | 0.15 | -44.19 | | | | | |
| OST3-11.5A | 0.28 | 0.25 | -31.94 | -32.76 | 1.16 | 2.70 | -35.58 | 11.49 ± 0.01 |
| OST3-11.5B | 0.32 | 0.24 | -33.58 | | | | | |
| OST3-11.3A | 0.12 | 0.14 | -39.55 | -40.01 | 0.64 | 2.70 | -42.63 | 11.30 ± 0.02 |
| OST3-11.3B | 0.15 | 0.11 | -40.46 | | | | | |





| FI sample | Water amount (μL) | Water content (μL/g) | δ²H (‰ VSMOW) measured | Mean δ²H (‰ VSMOW) | δ²H Std Dev | δ²H Error | Mean δ²H adjusted for IV (‰ VSMOW) | Age (kyr BP) |
|---|---|---|---|---|---|---|---|---|
| OST2-16.7 | 0.11 | 0.37 | -46.07 | -46.07 | n/a | 2.70 | -57.13 | 16.70 ± 0.07 |
| OST2-16.4A | 0.34 | 0.31 | -52.39 | | | | | |
| OST2-16.4B | 0.27 | 0.23 | -47.00 | -49.76 | 2.69 | 2.70 | -57.29 | 16.40 ± 0.05 |
| OST2-16.4C | 0.39 | 0.90 | -49.89 | | | | | |
| OST2-15.8A | 0.10 | 0.12 | -57.86 | -59.72 | 2.63 | 2.70 | -66.84 | 15.80 ± 0.07 |
| OST2-15.8B | 0.09 | 0.08 | -61.57 | | | | | |
| OST2-15.3 | 0.33 | 0.17 | -46.78 | -46.78 | n/a | 2.40 | -53.50 | 15.31 ± 0.08 |
| OST2-14.7 | 0.18 | 0.14 | -38.53 | -38.53 | n/a | 2.40 | -44.71 | 14.71 ± 0.18 |
| OST2-14.0A | 0.10 | 0.15 | -54.96 | -50.45 | 6.39 | 6.39 | -56.05 | 14.10 ± 0.09 |
| OST2-14.0B | 0.10 | 0.09 | -45.93 | | | | | |
| OST2-13.0A | 0.38 | 0.51 | -23.17 | | | | | |
| OST2-13.0B | 0.72 | 0.94 | -23.27 | -26.035 | 3.32 | 3.32 | -30.49 | 13.00 ± 0.08 |
| OST2-13.0C | 0.47 | 0.61 | -28.04 | | | | | |
| OST2-13.0D | 0.40 | 0.44 | -29.66 | | | | | |
| OST2-12.9A | 0.08 | 0.14 | -40.16 | -40.85 | 0.97 | 2.70 | -45.19 | 12.89 ± 0.07 |
| OST2-12.9B | 0.09 | 0.15 | -41.54 | | | | | |
| OST2-12.5A | 0.09 | 0.11 | -50.51 | -50.865 | 0.50 | 2.70 | -54.74 | 12.50 ± 0.10 |
| OST2-12.5B | 0.11 | 0.12 | -51.22 | | | | | |
| OST2-12.3A | 0.23 | 0.32 | -45.28 | -50.125 | 6.85 | 6.85 | -53.69 | 12.29 ± 0.10 |
| OST2-12.3B | 0.19 | 0.23 | -54.97 | | | | | |
| OST2-11.8 | 0.16 | 0.11 | -44.45 | -44.45 | n/a | 2.70 | -47.59 | 11.80 ± 0.03 |
| OST2-11.65A | 0.25 | 0.17 | -38.01 | -36.705 | 1.84 | 2.70 | -39.74 | 11.65 ± 0.02 |
| OST2-11.65B | 0.43 | 0.23 | -35.40 | | | | | |
| OST2-11.5A | 0.18 | 0.12 | -28.62 | -28.675 | 0.07 | 2.70 | -31.50 | 11.50 ± 0.01 |
| OST2-11.5B | 0.22 | 0.41 | -28.73 | | | | | |
| OST2-10.9A | 0.09 | 0.24 | -44.38 | -39.605 | 6.75 | 6.75 | -41.89 | 10.90 ± 0.08 |
| OST2-10.9B | 0.22 | 0.38 | -34.83 | | | | | |



**Table A2.** FI $\delta^2H$ measurements of Mendukilo samples. The $\delta^2H$ values were corrected for the ice-volume effect during the period covered by the Mendukilo speleothems. Each time span of each sample represents the duration covered by the respective calcite blocks sampled from the stalagmites used for the fluid inclusion measurements (without taking into account the age model uncertainty).

| FI sample | Water amount (µL) | Water content (µL/g) | $\delta^2H$ (‰ VSMOW) measured | Mean $\delta^2H$ (‰ VSMOW) | $\delta^2H$ Std Dev | $\delta^2H$ Error | Mean $\delta^2H$ adjusted for IV (‰ VSMOW) | Age (kyr BP) |
|---|---|---|---|---|---|---|---|---|
| Men2-botA | 0.61 | 0.29 | -37.67 | | | | | |
| Men2-botB | 0.54 | 0.33 | -37.62 | | | | | |
| Men2-botC | 0.18 | 0.11 | -44.43 | -40.40 | 2.91 | 2.91 | -44.85 | 12.90 ± 0.10 |
| Men2-botD | 0.28 | 0.21 | -40.35 | | | | | |
| Men2-botE | 0.30 | 0.18 | -41.94 | | | | | |
| Men2-0A | 0.18 | 0.19 | -48.93 | | | | | |
| Men2-0B | 0.32 | 0.21 | -47.88 | -47.75 | 1.24 | 2.70 | -51.98 | 12.78 ± 0.10 |
| Men2-0C | 0.30 | 0.29 | -46.44 | | | | | |
| Men2-5A | 0.16 | 0.27 | -55.39 | -57.33 | 2.74 | 2.74 | -61.31 | 12.65 ± 0.10 |
| Men2-5B | 0.14 | 0.13 | -59.26 | | | | | |
| Men2-10A | 0.33 | 0.20 | -47.04 | -48.12 | 1.53 | 2.70 | -52.00 | 12.51 ± 0.10 |
| Men2-10B | 0.12 | 0.11 | -49.20 | | | | | |
| Men2-17A | 0.07 | 0.08 | -46.23 | | | | | |
| Men2-17B | 0.21 | 0.19 | -52.85 | -47.74 | 3.81 | 3.81 | -51.40 | 12.32 ± 0.10 |
| Men2-17C | 0.07 | 0.09 | -48.00 | | | | | |
| Men2-17D | 0.35 | 0.21 | -43.86 | | | | | |
| Men2-22A | 0.13 | 0.11 | -44.45 | | | | | |
| Men2-22B | 0.15 | 0.11 | -51.36 | -46.58 | 4.14 | 4.14 | -50.14 | 12.21 ± 0.10 |
| Men2-22C | 0.23 | 0.12 | -43.93 | | | | | |
| Men2-27A | 0.22 | 0.22 | -46.64 | | | | | |
| Men2-27B | 0.26 | 0.29 | -47.66 | -48.10 | 1.89 | 2.70 | -51.45 | 12.08 ± 0.10 |
| Men2-27C | 0.22 | 0.22 | -50.87 | | | | | |
| Men2-27D | 0.12 | 0.10 | -47.24 | | | | | |
| Men2-35A | 0.71 | 0.62 | -43.10 | | | | | |
| Men2-35B | 0.40 | 0.29 | -45.77 | -45.31 | 1.89 | 2.70 | -48.45 | 11.83 ± 0.10 |
| Men2-35C | 0.62 | 0.55 | -44.76 | | | | | |
| Men2-35D | 0.16 | 0.14 | -47.62 | | | | | |
| Men2-43A | 0.15 | 0.15 | -46.48 | | | | | |
| Men2-43B | 0.23 | 0.26 | -38.55 | -44.28 | 3.83 | 3.83 | -47.10 | 11.59 ± 0.08 |
| Men2-43C | 0.14 | 0.15 | -46.28 | | | | | |
| Men2-43D | 0.14 | 0.18 | -45.79 | | | | | |
| Men2-47A | 0.25 | 0.36 | -48.00 | | | | | |
| Men2-47B | 0.26 | 0.36 | -52.12 | -50.72 | 2.35 | 2.70 | -53.54 | 11.50 ± 0.08 |
| Men2-47C | 0.13 | 0.18 | -52.03 | | | | | |
| Men2-48A | 0.15 | 0.17 | -35.09 | -33.49 | 2.26 | 2.70 | -36.32 | 11.48 ± 0.08 |
| Men2-48B | 0.51 | 0.59 | -31.89 | | | | | |





| FI sample | Water amount (μL) | Water content (μL/g) | δ²H (‰ VSMOW) measured | Mean δ²H (‰ VSMOW) | δ²H Std Dev | δ²H Error | Mean δ²H adjusted for IV (‰ VSMOW) | Age (kyr BP) |
|---|---|---|---|---|---|---|---|---|
| Men2-52A | 0.18 | 0.22 | -38.95 | | | | | |
| Men2-52B | 0.13 | 0.18 | -40.61 | -40.49 | 1.49 | 2.70 | -43.22 | 11.37 ± 0.08 |
| Men2-52C | 0.67 | 0.64 | -41.92 | | | | | |
| Men2-62A | 0.46 | 0.74 | -43.51 | | | | | |
| Men2-62B | 0.30 | 0.44 | -36.89 | -39.55 | 3.23 | 3.23 | -42.09 | 11.20 ± 0.08 |
| Men2-62C | 0.30 | 0.63 | -36.94 | | | | | |
| Men2-62D | 0.10 | 0.12 | -40.85 | | | | | |
| Men2-73A | 0.18 | 0.21 | -35.37 | -35.15 | 0.31 | 2.70 | -37.52 | 11.01 ± 0.06 |
| Men2-73B | 0.68 | 0.76 | -34.93 | | | | | |
| Men2-78A | 0.23 | 0.25 | -43.7 | | | | | |
| Men2-78B | 0.10 | 0.18 | -40.88 | -43.58 | 2.64 | 2.70 | -45.86 | 10.93 ± 0.06 |
| Men2-78C | 0.16 | 0.17 | -46.16 | | | | | |
| Men2-85A | 0.16 | 0.21 | -46.25 | | | | | |
| Men2-85B | 0.27 | 0.26 | -48.37 | -46.39 | 1.90 | 2.70 | -48.59 | 10.81 ± 0.06 |
| Men2-85C | 0.18 | 0.19 | -44.56 | | | | | |
| Men2-92A | 0.32 | 0.3 | -35.24 | | | | | |
| Men2-92B | 0.21 | 0.22 | -38.56 | -37.76 | 2.23 | 2.70 | -39.88 | 10.69 ± 0.06 |
| Men2-92C | 0.20 | 0.25 | -39.48 | | | | | |
| Men2-97A | 0.26 | 0.29 | -43.22 | | | | | |
| Men2-97B | 0.18 | 0.21 | -41.79 | -43.25 | 1.48 | 2.70 | -45.3 | 10.60 ± 0.06 |
| Men2-97C | 0.14 | 0.12 | -44.75 | | | | | |
| Men2-108A | 0.13 | 0.13 | -36.06 | -37.03 | 1.37 | 2.70 | -38.95 | 10.41 ± 0.06 |
| Men2-108B | 0.35 | 0.38 | -38 | | | | | |
| Men2-116A | 0.14 | 0.17 | -46.21 | | | | | |
| Men2-116B | 0.33 | 0.35 | -38.84 | -42.83 | 3.72 | 3.72 | -44.69 | 10.28 ± 0.06 |
| Men2-116C | 0.17 | 0.14 | -43.45 | | | | | |
| Men2-122A | 0.24 | 0.3 | -49.32 | | | | | |
| Men2-122B | 0.12 | 0.14 | -45.04 | -46.42 | 2.51 | 2.70 | -48.17 | 10.07 ± 0.06 |
| Men2-122C | 0.55 | 0.48 | -44.89 | | | | | |
| Men2-128A | 0.15 | 0.17 | -31.97 | | | | | |
| Men2-128B | 0.12 | 0.14 | -29.34 | -32.91 | 4.11 | 4.11 | -34.56 | 9.96 ± 0.08 |
| Men2-128C | 0.44 | 0.41 | -37.41 | | | | | |
| Men2-134A | 0.32 | 0.33 | -41.27 | | | | | |
| Men2-134B | 0.39 | 0.46 | -39.7 | -40.59 | 0.80 | 2.70 | -42.18 | 9.84 ± 0.08 |
| Men2-134C | 0.25 | 0.18 | -40.79 | | | | | |
| Men2-155A | 0.66 | 0.61 | -37.15 | | | | | |
| Men2-155B | 0.35 | 0.34 | -32.35 | -38.19 | 6.42 | 6.42 | -39.57 | 9.43 ± 0.08 |
| Men2-155C | 0.39 | 0.36 | -45.06 | | | | | |
| Men2-162A | 0.19 | 0.15 | -53.89 | | | | | |
| Men2-162B | 0.21 | 0.16 | -47.28 | -49.26 | 3.16 | 3.16 | -50.59 | 9.29 ± 0.08 |
| Men2-162C | 0.11 | 0.13 | -47.23 | | | | | |
| Men2-162D | 0.21 | 0.13 | -48.63 | | | | | |



| FI sample | Water amount (µL) | Water content (µL/g) | δ²H (‰ VSMOW) measured | Mean δ²H (‰ VSMOW) | δ²H Std Dev | δ²H Error | Mean δ²H adjusted for IV (‰ VSMOW) | Age (kyr BP) |
|---|---|---|---|---|---|---|---|---|
| Men2-177A | 0.33 | 0.32 | -39.30 | -39.17 | 0.18 | 2.70 | -40.38 | 9.01 ± 0.08 |
| Men2-177B | 0.35 | 0.23 | -39.05 | | | | | |
| Men2-185A | 0.10 | 0.10 | -44.21 | -44.13 | 3.42 | 3.42 | -45.25 | 8.85 ± 0.08 |
| Men2-185B | 0.19 | 0.10 | -47.50 | | | | | |
| Men2-185C | 0.21 | 0.12 | -40.67 | | | | | |
| Men2-192A | 0.14 | 0.16 | -42.95 | -38.32 | 4.25 | 4.25 | -39.41 | 8.72 ± 0.08 |
| Men2-192B | 0.17 | 0.16 | -34.61 | | | | | |
| Men2-192C | 0.38 | 0.19 | -37.39 | | | | | |
| Men2-200A | 0.38 | 0.35 | -40.42 | -42.33 | 1.69 | 2.70 | -43.35 | 8.57 ± 0.08 |
| Men2-200B | 0.37 | 0.24 | -42.93 | | | | | |
| Men2-200C | 0.22 | 0.17 | -43.63 | | | | | |
| Men2-207A | 0.11 | 0.09 | -43.67 | -43.23 | 0.62 | 2.70 | -44.23 | 8.43 ± 0.08 |
| Men2-207B | 0.19 | 0.15 | -42.79 | | | | | |
| Men2-215A | 0.15 | 0.12 | -48.65 | -52.55 | 3.96 | 3.96 | -53.51 | 8.28 ± 0.08 |
| Men2-215B | 0.18 | 0.14 | -56.10 | | | | | |
| Men2-215C | 0.25 | 0.15 | -55.81 | | | | | |
| Men2-215D | 0.25 | 0.18 | -49.63 | | | | | |
| Men2-251A | 0.23 | 0.16 | -42.11 | -39.36 | 2.39 | 2.70 | -40.15 | 7.54 ± 0.08 |
| Men2-251B | 0.46 | 0.35 | -38.20 | | | | | |
| Men2-251C | 0.17 | 0.09 | -37.78 | | | | | |
| Men2-267A | 0.13 | 0.12 | -43.70 | -49.38 | 4.04 | 4.04 | -50.11 | 7.20 ± 0.08 |
| Men2-267B | 0.16 | 0.10 | -52.75 | | | | | |
| Men2-267C | 0.16 | 0.10 | -51.67 | | | | | |
| Men2-267D | 0.20 | 0.13 | -49.38 | | | | | |
| Men2-282A | 0.15 | 0.11 | -44.16 | -45.59 | 1.72 | 2.70 | -46.27 | 6.88 ± 0.08 |
| Men2-282B | 0.38 | 0.27 | -45.10 | | | | | |
| Men2-282C | 0.20 | 0.11 | -47.50 | | | | | |
| Men2-295A | 0.19 | 0.15 | -47.52 | -46.54 | 3.14 | 3.14 | -47.18 | 6.59 ± 0.08 |
| Men2-295B | 0.22 | 0.21 | -42.13 | | | | | |
| Men2-295C | 0.16 | 0.12 | -46.96 | | | | | |
| Men2-295D | 0.17 | 0.13 | -49.54 | | | | | |
| Men2-301A | 0.11 | 0.08 | -49.69 | -49.93 | 1.25 | 2.70 | -50.53 | 6.47 ± 0.08 |
| Men2-301B | 0.21 | 0.14 | -48.81 | | | | | |
| Men2-301C | 0.14 | 0.08 | -51.28 | | | | | |
| Men2-307A | 0.08 | 0.05 | -43.49 | -44.21 | 1.84 | 2.70 | -44.8 | 6.34 ± 0.08 |
| Men2-307B | 0.16 | 0.11 | -46.3 | | | | | |
| Men2-307C | 0.16 | 0.11 | -42.84 | | | | | |
| Men2-310A | 0.20 | 0.18 | -49.79 | -46.10 | 5.06 | 5.06 | -46.69 | 6.28 ± 0.08 |
| Men2-310B | 0.25 | 0.21 | -40.32 | | | | | |
| Men2-310C | 0.18 | 0.12 | -48.18 | | | | | |
| Men2-312A | 0.38 | 0.25 | -45.11 | -44.22 | 0.83 | 2.70 | -44.79 | 6.20 ± 0.08 |
| Men2-312B | 0.52 | 0.35 | -44.08 | | | | | |
| Men2-312C | 0.33 | 0.34 | -43.46 | | | | | |



| FI sample | Water amount (μL) | Water content (μL/g) | δ²H (‰ VSMOW) measured | Mean δ²H (‰ VSMOW) | δ²H Std Dev | δ²H Error | Mean δ²H adjusted for IV (‰ VSMOW) | Age (kyr BP) |
|---|---|---|---|---|---|---|---|---|
| Men5-10A | 0.82 | 0.94 | -41.84 | | | | | |
| Men5-10B | 0.19 | 0.34 | -36.02 | -38.74 | 2.93 | 2.93 | -39.84 | 8.72 ± 0.06 |
| Men5-10C | 0.19 | 0.19 | -38.37 | | | | | |
| Men5-20A | 0.23 | 0.24 | -40.05 | | | | | |
| Men5-20B | 0.28 | 0.26 | -41.18 | -39.75 | 1.60 | 2.70 | -40.81 | 8.58 ± 0.06 |
| Men5-20C | 0.21 | 0.19 | -38.02 | | | | | |
| Men5-30A | 0.15 | 0.16 | -44.41 | | | | | |
| Men5-30B | 0.44 | 0.34 | -37.73 | -40.39 | 3.54 | 3.54 | -41.38 | 8.45 ± 0.06 |
| Men5-30C | 0.24 | 0.18 | -39.02 | | | | | |
| Men5-40A | 0.39 | 0.35 | -48.60 | | | | | |
| Men5-40B | 0.11 | 0.10 | -61.67 | -55.29 | 6.54 | 6.54 | -56.26 | 8.31 ± 0.06 |
| Men5-40C | 0.11 | 0.10 | -55.61 | | | | | |
| Men5-50A | 0.23 | 0.20 | -37.79 | | | | | |
| Men5-50B | 0.50 | 0.50 | -42.30 | -39.76 | 2.31 | 2.70 | -40.68 | 8.17 ± 0.06 |
| Men5-50C | 0.34 | 0.36 | -39.20 | | | | | |
| Men5-60A | 0.30 | 0.25 | -42.78 | | | | | |
| Men5-60B | 0.49 | 0.40 | -47.14 | -43.98 | 2.76 | 2.76 | -44.87 | 8.04 ± 0.06 |
| Men5-60C | 0.20 | 0.18 | -42.03 | | | | | |
| Men5-70A | 0.14 | 0.16 | -39.45 | | | | | |
| Men5-70B | 0.29 | 0.29 | -43.86 | -42.62 | 3.91 | 3.91 | -43.483 | 7.90 ± 0.06 |
| Men5-70C | 0.09 | 0.11 | -47.60 | | | | | |
| Men5-70D | 0.20 | 0.16 | -39.55 | | | | | |
| Men5-75A | 0.16 | 0.16 | -50.84 | | | | | |
| Men5-75B | 0.24 | 0.24 | -48.46 | -49.47 | 1.23 | 2.70 | -50.31 | 7.84 ± 0.06 |
| Men5-75C | 0.27 | 0.27 | -49.10 | | | | | |
| Men5-80A | 0.18 | 0.18 | -33.70 | | | | | |
| Men5-80B | 0.33 | 0.30 | -37.03 | -37.37 | 3.84 | 3.84 | -38.21 | 7.77 ± 0.06 |
| Men5-80C | 0.23 | 0.21 | -41.37 | | | | | |
| Men5-90A | 0.22 | 0.24 | -41.35 | | | | | |
| Men5-90B | 0.20 | 0.19 | -48.91 | -43.97 | 4.49 | 4.49 | -44.77 | 7.63 ± 0.06 |
| Men5-90C | 0.12 | 0.19 | -46.44 | | | | | |
| Men5-90D | 0.29 | 0.23 | -39.16 | | | | | |
| Men5-100A | 0.12 | 0.14 | -37.33 | | | | | |
| Men5-100B | 0.10 | 0.14 | -45.66 | -40.53 | 4.49 | 4.49 | -41.31 | 7.50 ± 0.06 |
| Men5-100C | 0.21 | 0.16 | -38.59 | | | | | |
| Men5-110A | 0.25 | 0.22 | -40.90 | | | | | |
| Men5-110B | 0.20 | 0.17 | -33.13 | -36.39 | 4.03 | 4.03 | -37.16 | 7.37 ± 0.06 |
| Men5-110C | 0.56 | 0.44 | -35.13 | | | | | |
| Men5-120A | 0.21 | 0.21 | -42.86 | | | | | |
| Men5-120B | 0.63 | 0.73 | -44.84 | -44.00 | 1.08 | 2.7 | -44.74 | 7.23 ± 0.06 |
| Men5-120C | 0.73 | 0.87 | -43.29 | | | | | |
| Men5-120D | 0.32 | 0.55 | -45.01 | | | | | |



| FI sample | Water amount (μL) | Water content (μL/g) | δ²H (‰ VSMOW) measured | Mean δ²H (‰ VSMOW) | δ²H Std Dev | δ²H Error | Mean δ²H adjusted for IV (‰ VSMOW) | Age (kyr BP) |
|---|---|---|---|---|---|---|---|---|
| Men5-130A | 0.19 | 0.16 | -42.10 | | | | | |
| Men5-130B | 0.34 | 0.27 | -47.22 | -43.79 | 2.97 | 2.97 | -44.51 | 7.08 ± 0.06 |
| Men5-130C | 0.21 | 0.22 | -42.05 | | | | | |
| Men5-140A | 0.18 | 0.23 | -49.92 | -48.32 | 2.26 | 2.7 | -49.01 | 6.93 ± 0.06 |
| Men5-140B | 0.20 | 0.23 | -46.72 | | | | | |
| Men5-150A | 0.26 | 0.26 | -42.23 | | | | | |
| Men5-150B | 0.30 | 0.23 | -46.99 | -43.88 | 2.70 | 2.70 | -44.53 | 6.75 ± 0.06 |
| Men5-150C | 0.21 | 0.19 | -42.41 | | | | | |
| Men5-160A | 0.35 | 0.32 | -40.50 | | | | | |
| Men5-160B | 0.47 | 0.39 | -45.46 | -42.37 | 2.69 | 2.70 | -43.01 | 6.58 ± 0.06 |
| Men5-160C | 0.30 | 0.21 | -41.16 | | | | | |
| Men5-170A | 0.22 | 0.22 | -34.99 | | | | | |
| Men5-170B | 0.27 | 0.27 | -37.43 | -38.44 | 2.85 | 2.85 | -39.04866731 | 6.45 ± 0.06 |
| Men5-170C | 0.20 | 0.41 | -39.83 | | | | | |
| Men5-170D | 0.19 | 0.14 | -41.52 | | | | | |
| Men5-180A | 0.39 | 0.3 | -44.03 | | | | | |
| Men5-180B | 0.3 | 0.3 | -40.73 | -42.33 | 1.65 | 2.70 | -42.92 | 6.31 ± 0.08 |
| Men5-180C | 0.45 | 0.34 | -42.22 | | | | | |
| Men5-190A | 0.21 | 0.19 | -36.51 | -37.98 | 2.08 | 2.7 | -38.5376 | 6.11 ± 0.08 |
| Men5-190B | 0.28 | 0.21 | -39.45 | | | | | |
| Men5-200A | 0.19 | 0.17 | -47.17 | | | | | |
| Men5-200B | 0.17 | 0.14 | -46.04 | -43.85 | 4.81 | 4.81 | -44.37 | 5.92 ± 0.08 |
| Men5-200C | 0.30 | 0.22 | -38.34 | | | | | |
| Men5-210A | 0.49 | 0.45 | -36.94 | | | | | |
| Men5-210B | 0.29 | 0.27 | -42.43 | -38.92 | 3.05 | 3.05 | -39.40 | 5.72 ± 0.08 |
| Men5-210C | 0.36 | 0.3 | -37.38 | | | | | |
| Men5-220A | 0.34 | 0.68 | -39.84 | | | | | |
| Men5-220B | 0.32 | 0.45 | -46.05 | -44.70 | 4.34 | 4.34 | -45.15 | 5.51 ± 0.08 |
| Men5-220C | 0.24 | 0.32 | -48.21 | | | | | |
| Men5-230A | 0.78 | 0.65 | -36.00 | | | | | |
| Men5-230B | 0.39 | 0.39 | -37.48 | -39.08 | 3.26 | 3.26 | -39.4999 | 5.31 ± 0.08 |
| Men5-230C | 0.26 | 0.42 | -39.30 | | | | | |
| Men5-230D | 0.18 | 0.20 | -43.53 | | | | | |
| Men5-240A | 0.28 | 0.23 | -45.74 | | | | | |
| Men5-240B | 0.12 | 0.12 | -45.76 | -46.09 | 2.82 | 2.82 | -46.4852 | 5.11 ± 0.08 |
| Men5-240C | 0.08 | 0.13 | -43.00 | | | | | |
| Men5-240D | 0.19 | 0.32 | -49.86 | | | | | |
| Men5-250A | 0.15 | 0.14 | -42.27 | | | | | |
| Men5-250B | 0.17 | 0.15 | -37.67 | -39.25 | 2.62 | 2.70 | -39.62 | 4.90 ± 0.08 |
| Men5-250C | 0.36 | 0.51 | -37.8 | | | | | |
| Men5-260A | 0.32 | 0.29 | -44.55 | -43.7 | 1.20 | 2.7 | -44.052 | 4.70 ± 0.08 |
| Men5-260B | 0.27 | 0.25 | -42.85 | | | | | |



| FI sample | Water amount (μL) | Water content (μL/g) | δ²H (‰ VSMOW) measured | Mean δ²H (‰ VSMOW) | δ²H Std Dev | δ²H Error | Mean δ²H adjusted for IV (‰ VSMOW) | Age (kyr BP) |
|---|---|---|---|---|---|---|---|---|
| Men5-280A | 0.3 | 0.49 | -48.24 | -48.94 | 0.98 | 2.7 | -49.2582 | 4.29 ± 0.08 |
| Men5-280B | 0.39 | 0.66 | -49.63 | | | | | |
| Men5-290A | 0.26 | 0.21 | -46.14 | -45.31 | 3.15 | 3.15 | -45.62 | 4.08 ± 0.08 |
| Men5-290B | 0.15 | 0.13 | -47.96 | | | | | |
| Men5-290C | 0.34 | 0.30 | -41.83 | | | | | |
| Men5-300A | 0.19 | 0.17 | -40.54 | -38.64 | 1.93 | 2.70 | -38.93 | 3.88 ± 0.08 |
| Men5-300B | 0.20 | 0.16 | -36.69 | | | | | |
| Men5-300C | 0.46 | 0.46 | -38.68 | | | | | |
| Men5-310A | 0.32 | 0.27 | -44.06 | -43.78 | 1.10 | 2.70 | -44.06 | 3.67 ± 0.08 |
| Men5-310B | 0.23 | 0.20 | -42.56 | | | | | |
| Men5-310C | 0.40 | 0.32 | -44.71 | | | | | |
| Men5-330A | 0.18 | 0.14 | -39.60 | -41.31 | 3.80 | 3.80 | -41.56 | 3.26 ± 0.08 |
| Men5-330B | 0.75 | 0.51 | -45.67 | | | | | |
| Men5-330C | 0.31 | 0.52 | -38.65 | | | | | |
| Men5-340A | 0.21 | 0.17 | -40.10 | -44.00 | 3.40 | 3.40 | -44.23 | 3.06 ± 0.08 |
| Men5-340B | 0.51 | 0.51 | -46.30 | | | | | |
| Men5-340C | 0.77 | 0.72 | -45.60 | | | | | |
| Men5-350A | 0.34 | 0.31 | -40.68 | -41.63 | 2.88 | 2.88 | -41.85 | 2.85 ± 0.08 |
| Men5-350B | 0.14 | 0.17 | -44.87 | | | | | |
| Men5-350C | 0.26 | 0.24 | -39.35 | | | | | |
| Men5-360A | 0.24 | 0.19 | -40.35 | -37.63 | 3.10 | 3.10 | -37.84 | 2.65 ± 0.08 |
| Men5-360B | 0.25 | 0.22 | -34.25 | | | | | |
| Men5-360C | 0.47 | 0.38 | -38.28 | | | | | |
| Men5-380A | 0.29 | 0.24 | -37.38 | -37.11 | 1.7 | 2.7 | -37.3101276 | 2.37 ± 0.06 |
| Men5-380B | 0.18 | 0.16 | -34.88 | | | | | |
| Men5-380C | 0.42 | 0.50 | -39.02 | | | | | |
| Men5-380D | 0.28 | 0.24 | -37.16 | | | | | |
| Men5-390A | 0.45 | 0.40 | -40.75 | -47.66 | 4.91 | 4.91 | -47.843 | 2.25 ± 0.04 |
| Men5-390B | 0.21 | 0.27 | -48.49 | | | | | |
| Men5-390C | 0.15 | 0.16 | -52.38 | | | | | |
| Men5-390D | 0.15 | 0.12 | -49.00 | | | | | |
| Men5-430A | 0.20 | 0.20 | -37.13 | -34.01 | 3.05 | 3.05 | -34.18 | 1.84 ± 0.04 |
| Men5-430B | 0.25 | 0.19 | -33.86 | | | | | |
| Men5-430C | 0.43 | 0.43 | -31.03 | | | | | |
| Men5-440A | 0.30 | 0.25 | -32.55 | -35.98 | 3.36 | 3.36 | -36.14 | 1.73 ± 0.04 |
| Men5-440B | 0.26 | 0.20 | -36.12 | | | | | |
| Men5-440C | 0.45 | 0.37 | -39.27 | | | | | |
| Men5-450A | 0.13 | 0.21 | -46.89 | -46.48 | 0.58 | 2.7 | -46.6384 | 1.63 ± 0.04 |
| Men5-450B | 0.21 | 0.24 | -46.07 | | | | | |
| Men5-460A | 0.39 | 0.34 | -39.2 | -38.00 | 2.28 | 2.70 | -38.15 | 1.53 ± 0.04 |
| Men5-460B | 0.15 | 0.15 | -35.37 | | | | | |
| Men5-460C | 0.50 | 0.40 | -39.42 | | | | | |




| FI sample | Water amount (μL) | Water content (μL/g) | δ²H (‰ VSMOW) measured | Mean δ²H (‰ VSMOW) | δ²H Std Dev | δ²H Error | Mean δ²H adjusted for IV (‰ VSMOW) | Age (kyr BP) |
|---|---|---|---|---|---|---|---|---|
| Men5-470A | 0.34 | 0.31 | -35.32 | | | | | |
| Men5-470B | 0.47 | 0.56 | -40.48 | -39.58 | 3.88 | 3.88 | -39.73 | 1.43 ± 0.04 |
| Men5-470C | 0.16 | 0.17 | -42.93 | | | | | |
| Men5-480A | 0.26 | 0.29 | -44.99 | | | | | |
| Men5-480B | 0.19 | 0.21 | -37.38 | -41.20 | 3.81 | 3.81 | -41.35 | 1.34 ± 0.04 |
| Men5-480C | 0.14 | 0.12 | -41.24 | | | | | |
| Men5-490A | 0.18 | 0.13 | -40.52 | | | | | |
| Men5-490B | 0.21 | 0.17 | -43.44 | -41.50 | 1.68 | 2.70 | -41.64 | 1.24 ± 0.04 |
| Men5-490C | 0.38 | 0.32 | -40.54 | | | | | |
| Men5-505A | 0.33 | 0.29 | -34.40 | | | | | |
| Men5-505B | 0.15 | 0.13 | -35.35 | -35.45 | 1.11 | 2.70 | -35.59 | 1.10 ± 0.04 |
| Men5-505C | 0.14 | 0.12 | -36.61 | | | | | |
| Men5-515A | 0.18 | 0.15 | -43.39 | | | | | |
| Men5-515B | 0.14 | 0.12 | -49.37 | -46.27 | 3.00 | 3.00 | -46.40 | 0.99 ± 0.04 |
| Men5-515C | 0.26 | 0.20 | -46.04 | | | | | |
| Men5-525A | 0.21 | 0.20 | -36.35 | | | | | |
| Men5-525B | 0.21 | 0.19 | -36.79 | -38.42 | 3.21 | 3.21 | -38.55 | 0.90 ± 0.04 |
| Men5-525C | 0.48 | 0.39 | -42.11 | | | | | |
| Men5-540A | 0.22 | 0.19 | -43.04 | | | | | |
| Men5-540B | 0.29 | 0.25 | -46.74 | -44.97 | 1.86 | 2.70 | -45.09 | 0.75 ± 0.04 |
| Men5-540C | 0.18 | 0.13 | -45.13 | | | | | |
| Men5-550A | 0.16 | 0.15 | -46.65 | | | | | |
| Men5-550B | 0.41 | 0.39 | -44.63 | -46.36 | 1.60 | 2.70 | -46.48 | 0.65 ± 0.04 |
| Men5-550C | 0.18 | 0.20 | -47.79 | | | | | |
| Men5-560A | 0.36 | 0.37 | -42.81 | | | | | |
| Men5-560B | 0.22 | 0.23 | -44.28 | -45.39 | 3.28 | 3.28 | -45.51 | 0.55 ± 0.04 |
| Men5-560C | 0.29 | 0.26 | -49.08 | | | | | |
| Men5-570A | 0.56 | 0.45 | -44.35 | | | | | |
| Men5-570B | 0.57 | 0.43 | -39.72 | -43.13 | 3.00 | 3.00 | -43.25 | 0.45 ± 0.04 |
| Men5-570C | 0.43 | 0.37 | -45.33 | | | | | |
| Men5-580A | 0.53 | 0.46 | -41.65 | | | | | |
| Men5-580B | 0.47 | 0.50 | -45.46 | -43.02 | 2.12 | 2.70 | -43.13 | 0.35 ± 0.04 |
| Men5-580C | 0.48 | 0.45 | -41.95 | | | | | |
| Men5-590A | 0.15 | 0.13 | -50.41 | | | | | |
| Men5-590B | 0.17 | 0.15 | -46.03 | -46.88 | 3.19 | 3.19 | -46.99 | 0.25 ± 0.04 |
| Men5-590C | 0.56 | 0.59 | -44.20 | | | | | |
| Men5-600A | 0.26 | 0.27 | -47.86 | | | | | |
| Men5-600B | 0.45 | 0.54 | -41.08 | -43.66 | 3.67 | 3.67 | -43.77 | 0.15 ± 0.04 |
| Men5-600C | 0.54 | 0.47 | -42.05 | | | | | |



**Table A3.** Paleotemperatures obtained from $\delta^2H_{FI}$ data using the OM-FIT transfer function.

| Sample (stratigraphic order) | Age (kyr BP) | Mean $\delta^2H$ adjusted for IV (‰ VSMOW) | $\delta^2H$ corrected error ± (‰) | Temp. (°C) OM-FIT | Error ± (°C) |
|---|---|---|---|---|---|
| OST2-16.7 | 16.70 ± 0.07 | -57.13 | 2.70 | 8.14 | 2.05 |
| OST2-16.4 | 16.40 ± 0.05 | -57.29 | 2.70 | 8.11 | 2.05 |
| OST1-16.1 | 16.06 ± 0.06 | -57.08 | 2.70 | 8.15 | 2.05 |
| OST2-15.8 | 15.80 ± 0.07 | -66.84 | 2.70 | 5.94 | 2.05 |
| OST2-15.3 | 15.31 ± 0.08 | -53.5 | 2.70 | 8.97 | 2.05 |
| OST1-15.2 | 15.16 ± 0.05 | -57.98 | 3.38 | 7.95 | 2.07 |
| OST2-14.7 | 14.78 ± 0.18 | -44.71 | 2.70 | 10.97 | 2.05 |
| OST1-14.6 | 14.57 ± 0.05 | -31.36 | 2.70 | 14.00 | 2.05 |
| OST3-14.3 | 14.30 ± 0.09 | -31.83 | 2.70 | 13.89 | 2.05 |
| OST1-14.2 | 14.20 ± 0.02 | -49.09 | 2.70 | 9.97 | 2.05 |
| OST3-14.1 | 14.11 ± 0.09 | -40.89 | 2.70 | 11.83 | 2.05 |
| OST2-14.0 | 14.10 ± 0.09 | -56.05 | 6.39 | 8.39 | 2.89 |
| OST3-13.5 | 13.50 ± 0.09 | -39.90 | 2.70 | 12.06 | 2.05 |
| OST2-13.0 | 13.00 ± 0.08 | -30.49 | 3.32 | 14.20 | 2.19 |
| Men2-bot | 12.90 ± 0.10 | -44.85 | 2.91 | 11.72 | 1.76 |
| OST2-12.9 | 12.89 ± 0.07 | -45.19 | 2.70 | 10.86 | 2.05 |
| OST3-12.8 | 12.80 ± 0.08 | -47.98 | 2.70 | 10.22 | 2.05 |
| Men2-0 | 12.78 ± 0.10 | -51.98 | 2.70 | 10.11 | 1.76 |
| Men2-5 | 12.65 ± 0.10 | -61.31 | 2.74 | 7.98 | 1.77 |
| Men2-10 | 12.51 ± 0.10 | -52.00 | 2.70 | 10.10 | 1.76 |
| OST2-12.5 | 12.50 ± 0.10 | -54.74 | 2.70 | 8.69 | 2.05 |
| Men2-17 | 12.32 ± 0.10 | -51.40 | 3.81 | 10.24 | 2.02 |
| OST2-12.3 | 12.29 ± 0.10 | -53.69 | 6.85 | 8.93 | 3.00 |
| Men2-22 | 12.21 ± 0.10 | -50.14 | 4.14 | 10.52 | 2.09 |
| Men2-27 | 12.08 ± 0.10 | -51.45 | 2.70 | 10.22 | 1.76 |
| Men2-35 | 11.83 ± 0.10 | -48.45 | 2.70 | 10.91 | 1.76 |
| OST2-11.8 | 11.80 ± 0.03 | -47.59 | 2.70 | 10.31 | 2.05 |
| OST3-11.7 | 11.67 ± 0.02 | -27.92 | 2.70 | 14.78 | 2.05 |
| OST2-11.65 | 11.65 ± 0.02 | -39.74 | 2.70 | 12.10 | 2.05 |
| OST3-11.6 | 11.60 ± 0.02 | -42.6 | 6.39 | 11.45 | 2.89 |
| Men2-43 | 11.59 ± 0.08 | -47.10 | 3.83 | 11.21 | 2.02 |
| Men2-47 | 11.50 ± 0.08 | -53.54 | 2.70 | 9.75 | 1.76 |
| OST2-11.5 | 11.50 ± 0.01 | -31.5 | 2.70 | 13.97 | 2.05 |
| Men2-48 | 11.48 ± 0.08 | -36.32 | 2.70 | 13.66 | 1.76 |
| Men2-52 | 11.37 ± 0.08 | -43.22 | 2.70 | 12.10 | 1.76 |
| OST3-11.3 | 11.30 ± 0.02 | -42.63 | 2.70 | 11.44 | 2.05 |
| Men2-62 | 11.20 ± 0.08 | -42.09 | 3.23 | 12.35 | 1.88 |
| Men2-73 | 11.01 ± 0.06 | -37.52 | 2.70 | 13.39 | 1.76 |
| OST1-10.9 | 10.95 ± 0.20 | -28.86 | 3.82 | 14.57 | 2.31 |
| Men2-78 | 10.93 ± 0.06 | -45.86 | 2.70 | 11.50 | 1.75 |



| | | | | | |
|---|---|---|---|---|---|
| OST2-10.9 | 10.90 ± 0.08 | -41.89 | 6.75 | 11.61 | 2.97 |
| Men2-85 | 10.81 ± 0.06 | -48.59 | 2.70 | 10.87 | 1.76 |
| Men2-92 | 10.69 ± 0.06 | -39.88 | 2.70 | 12.85 | 1.76 |
| Men2-97 | 10.60 ± 0.06 | -45.30 | 2.70 | 11.62 | 1.76 |
| Men2-108 | 10.41 ± 0.06 | -38.95 | 2.70 | 13.07 | 1.76 |
| Men2-116 | 10.28 ± 0.06 | -44.69 | 3.72 | 11.76 | 2.00 |
| Men2-122 | 10.07 ± 0.06 | -48.17 | 2.70 | 10.97 | 1.76 |
| Men2-128 | 9.96 ± 0.08 | -34.56 | 4.11 | 14.06 | 2.09 |
| Men2-134 | 9.84 ± 0.08 | -42.18 | 2.70 | 12.33 | 1.76 |
| Men2-155 | 9.43 ± 0.08 | -39.57 | 6.42 | 12.92 | 2.61 |
| Men2-162 | 9.29 ± 0.08 | -50.59 | 3.16 | 10.42 | 1.87 |
| Men2-177 | 9.01 ± 0.08 | -40.38 | 2.70 | 12.74 | 1.76 |
| Men2-185 | 8.85 ± 0.08 | -45.25 | 3.42 | 11.63 | 1.93 |
| Men5-10 | 8.72 ± 0.06 | -39.84 | 2.93 | 12.86 | 1.82 |
| Men5-20 | 8.58 ± 0.06 | -40.81 | 2.70 | 12.64 | 1.76 |
| Men2-200 | 8.57 ± 0.08 | -43.35 | 2.70 | 12.07 | 1.76 |
| Men5-30 | 8.45 ± 0.06 | -41.38 | 3.54 | 12.51 | 1.96 |
| Men2-207 | 8.43 ± 0.08 | -44.23 | 2.70 | 11.87 | 1.76 |
| Men5-40 | 8.31 ± 0.06 | -56.26 | 6.54 | 9.13 | 2.64 |
| Men2-215 | 8.28 ± 0.08 | -53.51 | 3.96 | 9.76 | 2.05 |
| Men5-50 | 8.17 ± 0.06 | -40.68 | 2.70 | 12.67 | 1.76 |
| Men5-60 | 8.04 ± 0.06 | -44.87 | 2.76 | 11.72 | 1.78 |
| Men5-70 | 7.90 ± 0.06 | -43.48 | 3.91 | 12.04 | 2.04 |
| Men5-75 | 7.84 ± 0.06 | -50.31 | 2.70 | 10.48 | 1.76 |
| Men5-80 | 7.77 ± 0.06 | -38.21 | 3.84 | 13.23 | 2.02 |
| Men5-90 | 7.63 ± 0.06 | -44.77 | 4.49 | 11.74 | 2.17 |
| Men2-251 | 7.54 ± 0.08 | -40.15 | 2.70 | 12.79 | 1.76 |
| Men5-100 | 7.50 ± 0.06 | -41.31 | 4.49 | 12.53 | 2.17 |
| Men5-110 | 7.37 ± 0.06 | -37.16 | 4.03 | 13.47 | 2.07 |
| Men5-120 | 7.23 ± 0.06 | -44.74 | 2.70 | 11.75 | 1.76 |
| Men2-267 | 7.20 ± 0.08 | -50.11 | 4.04 | 10.53 | 2.07 |
| Men5-130 | 7.08 ± 0.06 | -44.51 | 2.97 | 11.80 | 1.83 |
| Men5-140 | 6.93 ± 0.06 | -49.01 | 2.70 | 10.78 | 1.76 |
| Men2-282 | 6.88 ± 0.08 | -46.27 | 2.70 | 11.40 | 1.76 |
| Men5-150 | 6.75 ± 0.06 | -44.53 | 2.70 | 11.80 | 1.76 |
| Men2-295 | 6.59 ± 0.08 | -47.18 | 3.14 | 11.20 | 1.86 |
| Men5-160 | 6.58 ± 0.06 | -43.01 | 2.70 | 12.14 | 1.76 |
| Men2-301 | 6.47 ± 0.08 | -50.53 | 2.70 | 10.43 | 1.76 |
| Men5-170 | 6.45 ± 0.06 | -39.05 | 2.85 | 13.04 | 1.80 |
| Men2-307 | 6.34 ± 0.08 | -44.80 | 2.70 | 11.74 | 1.76 |
| Men5-180 | 6.31 ± 0.08 | -42.92 | 2.70 | 12.16 | 1.76 |
| Men2-310 | 6.28 ± 0.08 | -46.69 | 5.06 | 11.31 | 2.30 |
| Men2-312 | 6.20 ± 0.08 | -44.79 | 2.70 | 11.74 | 1.76 |
| Men5-190 | 6.11 ± 0.08 | -38.54 | 2.70 | 13.16 | 1.76 |
| Men5-200 | 5.92 ± 0.08 | -44.37 | 4.81 | 11.83 | 2.24 |
| Men5-210 | 5.72 ± 0.08 | -39.40 | 3.05 | 12.96 | 1.84 |
| Men5-220 | 5.51 ± 0.08 | -45.15 | 4.34 | 11.66 | 2.14 |





| | | | | | |
|---|---|---|---|---|---|
| Men5-230 | 5.31 ± 0.08 | -39.50 | 3.26 | 12.94 | 1.89 |
| Men5-240 | 5.11 ± 0.08 | -46.49 | 2.82 | 11.35 | 1.79 |
| Men5-250 | 4.90 ± 0.08 | -39.62 | 2.70 | 12.91 | 1.76 |
| Men5-260 | 4.70 ± 0.08 | -44.05 | 2.70 | 11.91 | 1.76 |
| Men5-280 | 4.29 ± 0.08 | -49.26 | 2.70 | 10.72 | 1.76 |
| Men5-290 | 4.08 ± 0.08 | -45.62 | 3.15 | 11.55 | 1.87 |
| Men5-300 | 3.88 ± 0.08 | -38.93 | 2.70 | 13.07 | 1.76 |
| Men5-310 | 3.67 ± 0.08 | -44.06 | 2.70 | 11.91 | 1.76 |
| Men5-330 | 3.26 ± 0.08 | -41.56 | 3.80 | 12.47 | 2.02 |
| Men5-340 | 3.06 ± 0.08 | -44.23 | 3.40 | 11.86 | 1.92 |
| Men5-350 | 2.85 ± 0.08 | -41.85 | 2.88 | 12.41 | 1.80 |
| Men5-360 | 2.65 ± 0.08 | -37.84 | 3.10 | 13.32 | 1.86 |
| Men5-380 | 2.37 ± 0.06 | -37.31 | 2.70 | 13.44 | 1.76 |
| Men5-390 | 2.25 ± 0.04 | -47.84 | 4.91 | 11.04 | 2.27 |
| Men5-430 | 1.84 ± 0.04 | -34.18 | 3.05 | 14.15 | 1.84 |
| Men5-440 | 1.73 ± 0.04 | -36.14 | 3.36 | 13.70 | 1.91 |
| Men5-450 | 1.63 ± 0.04 | -46.64 | 2.70 | 11.32 | 1.76 |
| Men5-460 | 1.53 ± 0.04 | -38.15 | 2.70 | 13.25 | 1.76 |
| Men5-470 | 1.43 ± 0.04 | -39.73 | 3.88 | 12.89 | 2.03 |
| Men5-480 | 1.34 ± 0.04 | -41.35 | 3.81 | 12.52 | 2.01 |
| Men5-490 | 1.24 ± 0.04 | -41.64 | 2.70 | 12.45 | 1.76 |
| Men5-505 | 1.10 ± 0.04 | -35.59 | 2.70 | 13.83 | 1.76 |
| Men5-515 | 0.99 ± 0.04 | -46.40 | 3.00 | 11.37 | 1.83 |
| Men5-525 | 0.90 ± 0.04 | -38.55 | 3.21 | 13.16 | 1.88 |
| Men5-540 | 0.75 ± 0.04 | -45.09 | 2.70 | 11.67 | 1.76 |
| Men5-550 | 0.65 ± 0.04 | -46.48 | 2.70 | 11.36 | 1.76 |
| Men5-560 | 0.55 ± 0.04 | -45.51 | 3.28 | 11.58 | 1.90 |
| Men5-570 | 0.45 ± 0.04 | -43.25 | 3.00 | 12.09 | 1.83 |
| Men5-580 | 0.35 ± 0.04 | -43.13 | 2.70 | 12.12 | 1.76 |
| Men5-590 | 0.25 ± 0.04 | -46.99 | 3.19 | 11.24 | 1.88 |
| Men5-600 | 0.15 ± 0.04 | -43.77 | 3.67 | 11.97 | 1.98 |



## DATA AVAILABILITY

The speleothem $\delta^{18}O$ data that support the findings of this study are available as a download excel file in the Supplement and all the fluid inclusion data will later be integrated in the SISAL database.

## AUTHOR CONTRIBUTIONS

J.B.W, A.M., M.B., C.P.M., M.A., contributed to design this research project. J.B.W., C.S., Y.D., E.I., I.C., provided the isotopic data. J.B.W., C.P.M., L.R.E., H.C., provided the chronological data. J.B.W., A.M., E.I., provided the thin sections and/or contributed in the petrographic characterization. J.B.W., A.M., M.B., C.P.M., M.A., R.J., E.I., helped during field work. All authors contributed to the writing of the manuscript.

## COMPETING INTEREST

The contact author has declared that none of the authors has any competing interests.

## ACKNOWLEDGEMENTS

We are grateful to the guides and workers of the Mendukilo cave, M. Larburu and A. Govillar, for helping with the monitoring work in the cave and preserving the sampling points. We are also grateful to M. Wimmer for support during lab work at Innsbruck University, and to all people who helped during field work in the Ostolo cave (I. Altzuri and K. Sanchez). We would like to acknowledge the use of Servicio de Apoyo a la Investigacion, Zaragoza and the staff of the IsoTOPIK laboratory at University of Burgos. I. Cacho thanks the Catalan Institution for Research and Advanced Studies (ICREA) academia program from the Generalitat de Catalunya.

## FINANCIAL SUPPORT

This research has been supported by the Spanish Agencia Estatal de Investigación (AEI) (grant nos. PID2019–106050RB-I00 (PYCACHU) and PID2022-139101OB-I00 (TEMPURA)). We acknowledge support of the publication fee by the CSIC Open Access Publication Support Initiative through its Unit of Information Resources for Research (URICI).



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



**FIGURE CAPTIONS**

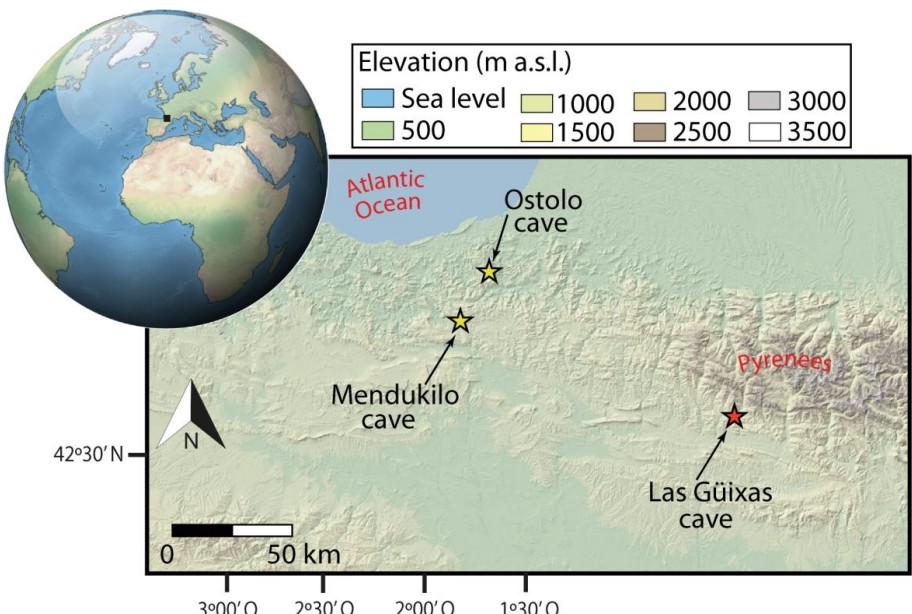

**Figure 1.** Location of the study area in Northern Spain. Yellow stars indicate the locations of the two studied caves, while the red star marks the site where the isotopic composition of rainfall was monitored (Las Güixas tourist cave in Villanúa).





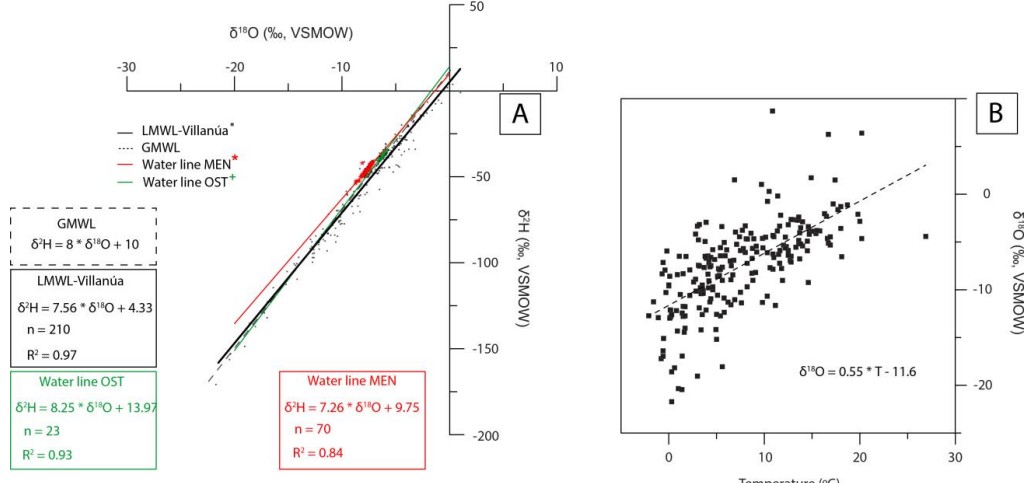

**Figure 2.** A) $\delta^2$H and $\delta^{18}$O values of precipitation events at Villanúa (black dots) along with the Local Meteoric Water Line (LMWL; black line). Samples that experienced evaporation prior to sampling and outliers were excluded (Giménez et al., 2021). The Global Meteoric Water Line (GMWL; dashed line; Rozanski et al., 1993) and the drip water lines of Mendukilo (MEN; red line) and Ostolo (OST; green line) are also represented. B) $\delta^{18}$O values of precipitation events and their respective temperature at Villanúa (Giménez et al., 2021). The dashed line represents the linear regression of precipitation isotope and temperature data.



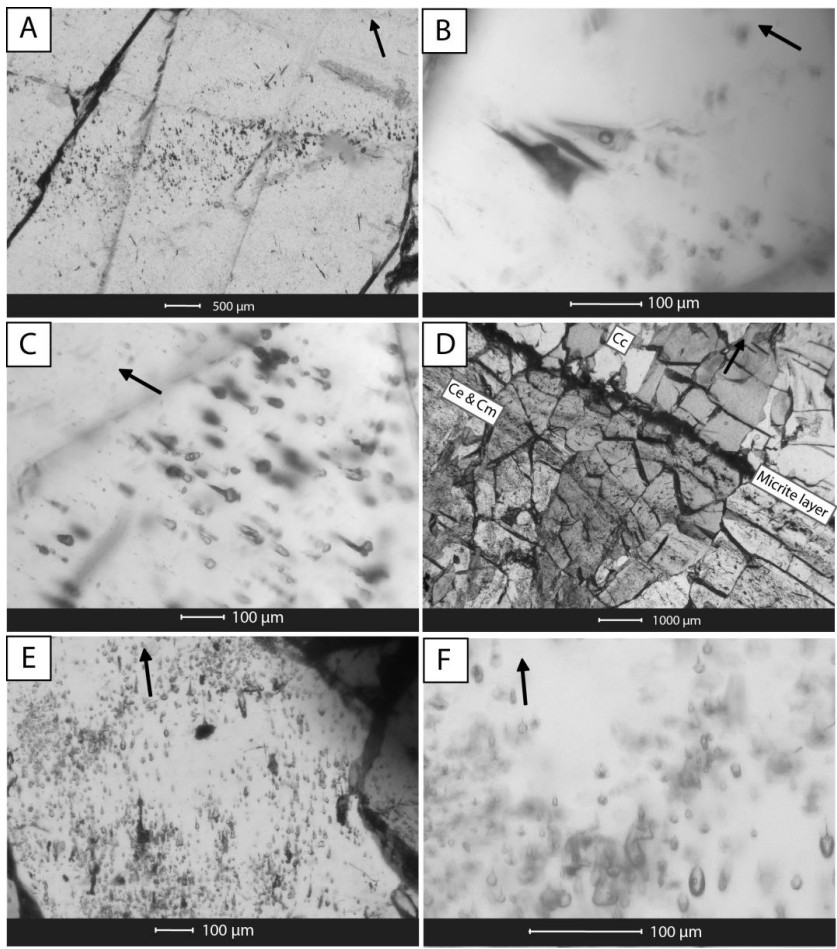

**Figure 3.** FI assemblages in the studied stalagmites. A) Primary FI throughout the growth layers in stalagmite MEN-5. B) Inter-crystalline FI in stalagmite MEN-2. C) Intra-crystalline FI in stalagmite MEN-5. D) Primary intra and inter-crystalline FI in stalagmite OST2, more frequently found in porous areas or associated with elongated (Ce) and/or microcrystalline (Cm) fabrics than with a tightly packed columnar fabric (Cc). E) FI in stalagmite OST2 are mostly intra-crystalline and does not necessarily align with the growth layers. F) Pyriform and rounded intra-crystalline small FI in stalagmite OST2. Black arrows indicate the speleothem growth direction.



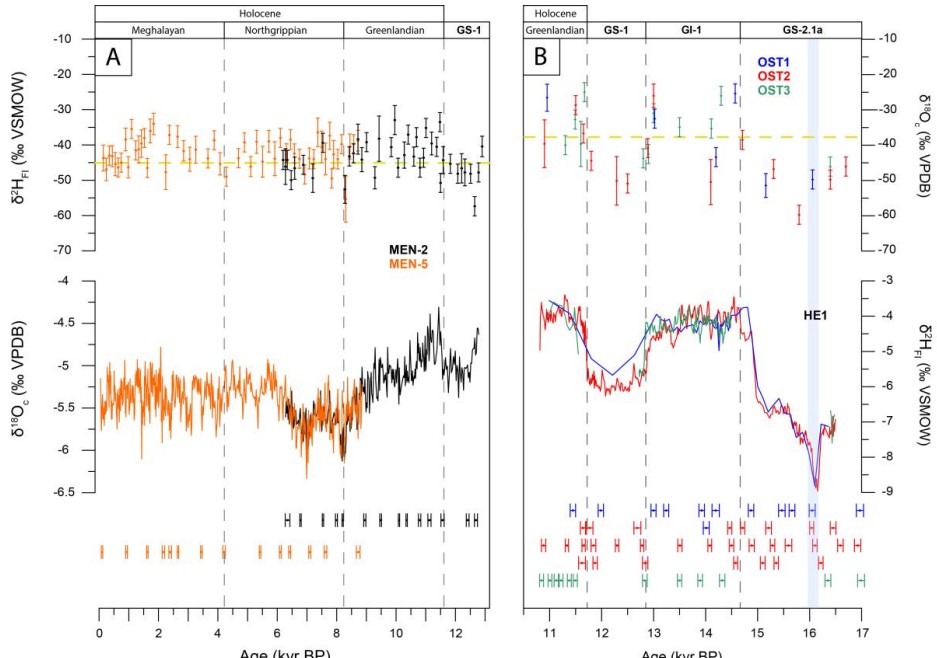

**Figure 4.** A) $\delta^2$H of FI water ($\delta^2$H$_{FI}$) and $\delta^{18}$O of calcite ($\delta^{18}$O$_c$) of Mendukilo (MEN-5 in orange and MEN-2 in black) and B) Ostolo stalagmites (OST1 in blue, OST2 in red, and OST3 in green). $\delta^2$H$_{FI}$ values are corrected for the ice-volume effect (Bintanja et al., 2005) with vertical error bars representing isotope measurements errors and 1$\sigma$ from repeated measurements. The yellow dashed line in the upper graphs of each panel indicates the annual mean $\delta^2$H value in drip water for each cave. Modeled U/Th ages with 2$\sigma$ error bars for stalagmites from each cave are shown at the bottom. Heinrich event 1 (HE1) recorded in the Ostolo isotope record (Bernal-Wormull et al., 2021) is highlighted by a light blue bar.



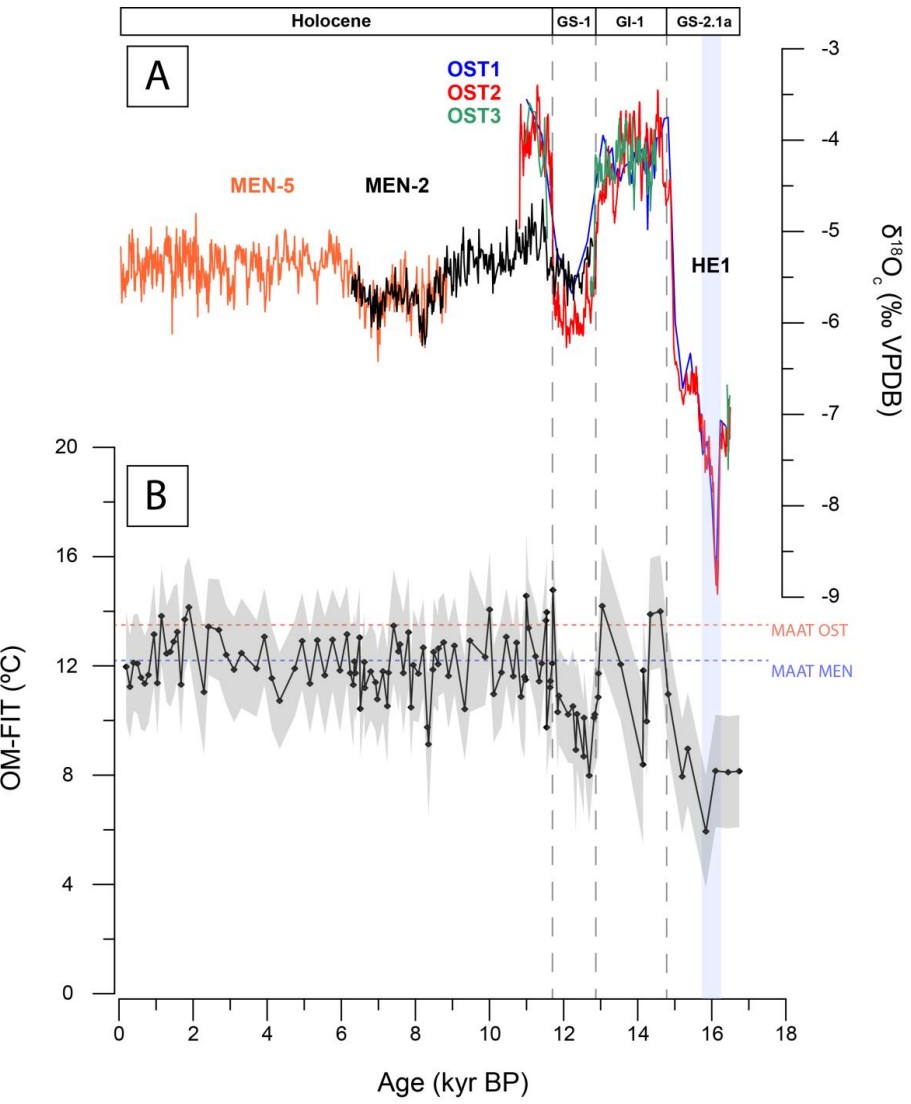

**Figure 5.** A) δ¹⁸Oc records from Mendukilo and Ostolo stalagmites, compared to B) the
OM-FIT paleotemperature reconstruction (bottom). Heinrich event 1 (HE1) is highlighted
by a light blue bar. The MAAT outside the two caves is shown by dashed horizontal lines.





**Figure 6.** Paleotemperature reconstructions over the last 16.5 kyr BP, spanning from Greenland to SW Europe, along with speleothem $\delta^{18}O$ records from the Iberian Peninsula. A) NGRIP $\delta^{18}O$ (gray solid line; Rasmussen et al., 2014) and Greenland temperature reconstruction (black solid line; Kindler et al., 2014). B) Milandre cave FI temperature record (MC-FIT) from NW Switzerland (Affolter et al., 2019). C) July temperature inferred from chironomids at Basa de la Mora Lake (Tarrats et al., 2018). D) Stacked and spliced chironomid-inferred July temperature record from SW Europe (Heiri et al., 2014b). E) Ostolo and Mendukilo FI temperature record (OM-FIT; yellow star; this study). F) $\delta^{18}Oc$ records from Mendukilo and Ostolo (Bernal-Wormull et al., 2021, 2023). G) LV5 $\delta^{18}O$ record from Kaite Cave (northern Iberia; Domínguez-Villar et al., 2017). H) GAR-01 $\delta^{18}O$ record from La Garma Cave (northern Iberia; Baldini et al., 2019). I) SIR-14 $\delta^{18}O$ record from El Soplao Cave (northern Iberia; Kilhavn et al., 2022). Key abrupt climate events (Heinrich 1 [HE1], 9.3 kyr and the 8.2 kyr events) and Greenland stadials (GS-1 and GS-2.1a) are highlighted by a light blue bar. The gray envelope around the solid lines in B), C), D) and E) show the uncertainties.