# Peer review of "constrained by speleothem fluid inclusion water isotopes"

_EGUsphere, 2024_

## Author Response (AR2)

**Referee #1 comments (14.12.2024)**

The manuscript presents a massive dataset of $\delta^2$H values of inclusion-hosted water for five speleothems, whose age segments are overlapping. The stable isotope composition records of the speleothems fit each other, proving that the signals are reproducible. The speleothems collectively cover the last 16.5 ky, their age precisions are very good. The paper is written clearly in most parts. All these together would support publication in a well known journal like Climate of the Past. The datasets, their descriptions, and the comparisons with independent records are all fine. However, the reviewer misses the discussion of consequences, the explanations how and why these data modify the knowledge of late Pleistocene and Holocene climate conditions and their governing processes in Iberia and the wider region. Both the Abstract and the Conclusions sections are confined to the descriptions of analytical results, the Discussions contain comparisons with other paleoclimate records, but not more. What is the significance of temperature differences between GS and GI periods and the Holocene? I suggest a major revision to address the governing processes of climate changes, most probably initiated in the North Atlantic.

Dear editor,

We are grateful for the constructive comments and helpful suggestions by the reviewer. Below is a point-by-point response to the major and minor comments and questions by **Referee #1:**

We agree with the reviewer that the abstract and conclusions of the manuscript do not mention the real consequences of the governing processes of climate changes during the period between GS and GI throughout the last deglaciation and the Holocene. Both sections lack a global perspective and we have modified them regarding the governing processes of climate changes, which were primarily initiated in the North Atlantic.

ABSTRACT: In the Northern Hemisphere, the last 16.5 kyr were characterized by abrupt temperature transitions during stadials, interstadials, and the onset of the Holocene. These changes are closely linked to large-scale variations in the extent of continental ice-sheets, greenhouse gas concentrations, and ocean circulation. Speleothems and their fluid inclusions serve as valuable proxies, offering high-resolution chronologies and quantitative records of past temperature changes for understanding global and regional climate mechanisms in the past. Here, we present a record based on five speleothems from two caves on the northeastern Iberian Peninsula (Ostolo and Medukilo caves). Using hydrogen isotopic composition of fluid inclusions and rainfall samples, we developed a $\delta^2$H/T transfer function in order to reconstruct regional temperatures over the past 16.5 kyr (Ostolo-Mendukilo Fluid Inclusion Temperature record [OM-FIT]). Our novel findings reveal abrupt temperature changes in SW Europe during the last deglaciation and early Holocene, at millennial and centennial scales, anchored by a precise chronology. At the onset of Greenland Interstadial 1, the OM-FIT records shows an increase of 6.7 ± 2.8 °C relative to the cold conditions of the preceding Greenland Stadial 2.1a. During the early phase of Greenland Stadial 1, OM-FIT records a temperature decline of 6.1 ± 2.8 °C. The end of this cold phase and the onset of the Holocene are marked by a rapid warming of about 5 °C, reaching a maximum at 11.66 ± 0.03 kyr BP. The OM-FIT record also exhibits abrupt events during the Holocene (e.g., the 8.2 kyr event), which are also reflected in the $\delta^{18}$O values of the calcite. These abrupt temperature changes during the last deglaciation and the Holocene correspond to variations seen in paleotemperature records across Europe and in Greenland ice cores. This clearly illustrates the influence of changes in the Atlantic

Meridional Overturning Circulation, driven by subarctic freshening, on the climate of southern Europe.

CONCLUSION: The Ostolo and Mendukilo speleothems provide a replicated and precisely dated record of paleotemperature in NE Iberia for the past 16.5 kyr BP. The OM-FIT record contributes novel, non-biogenic evidence of rapid temperature transitions and emphasize the significant connections between abrupt air temperature shifts in this Southwestern European terrestrial record and changes in deep-water ocean circulation and meltwater input during the last deglaciation and the Greenlandian period (on millennial and/or centennial time scales). Our findings indicate temperatures for GS-2.1a up to 6.7 ± 2.8 °C lower than those for GI-1 and present-day conditions, and constrain the regional response of HE-1 between 16.2 and 15.8 kyr BP with meltwater contributions from the North Atlantic. The sharp rise in temperatures during the GS-2.1a/GI-1 transition was quantitatively comparable to other records from SW Europe. Temperatures during GI-1 were equivalent to those of the Holocene, with a minimum observed at 14.1 ± 0.1 kyr BP during GI-1d. All these shifts and noticeable cooling events observed in the OM-FIT were not only synchronous with climate variations associated with subarctic freshening documented by other records of Europe, but were also synchronous with glacier advances in the Pyrenees during GS-2.1a and GI-1d. The rapid temperature changes at early GS-1 and the onset of the Holocene recorded by OM-FIT are consistent with to those reported from other parts of Europe and also confirm quantitatively equally high temperatures for both GI-1 and the onset of the Holocene. Neither $\delta^{18}O_C$ nor OM-FIT reveal significant millennial-scale changes during the Holocene. The 8.2 kyr event is recorded between 8.29 and 8.10 ± 0.04 kyr in the $\delta^{18}O_C$ record, centered at 8.29 ± 0.07 kyr in the OM-FIT record, synchronous with Greenland ice-core data and well-dated records from central and SW Europe. Future quantitative temperature studies using well-dated speleothems from SW Europe are necessary to corroborate the chronology and better delineate the drastic centennial-scale temperature changes during the last deglaciation where the OM-FIT only has a few data points (e.g., HE1 and GI-1d). On the other hand, the intricate interactions of the coupled ocean-atmosphere system during the Holocene, particularly in its later stages and on a millennial scale, underline the need for further research into the driving mechanisms and feedbacks that are crucial in influencing natural temperature variability during the recent interglacial in southern Europe.

Specific comments:

Fig. 4. B axis titles are mixed.

We fixed the titles of the axis in Fig. 4B.

line 113: it is not clear how the isotopically equilibrium precipitation of carbonate would affect the stable hydrogen isotope composition of inclusion-hosted water. Please explain.

We thank the reviewer for his comments in this section where we were really referring only to the oxygen isotopes and how the isotopically equilibrium precipitation of carbonate would affect them. The $\delta^2H_{FI}$ values are the most suitable temperature proxy as they witness surface air temperature. In the manuscript this is not clear to the reader since it is not specified that only oxygen isotopes are being discussed in this particular case of non-equilibrium conditions. The text in line 113 is now corrected accordingly to make it clear that this effect can be relevant in the $\delta^{18}O_{FI}$ values, giving a brief explanation of the origins of this imbalance, but without arguing too much, since as explained later in the article, in this work only hydrogen values are used to calculate paleotemperatures.

Line 111: For the FI water isotope thermometry method to yield reliable results, four aspects must be considered: (i) FIs must be of primary origin, well-sealed, and sufficiently abundant; (ii) the choice of the transfer function converting the hydrogen and/or oxygen isotope signal ($\delta^2H_{FI}$, $\delta^{18}O_{FI}$) into temperature may bias temperature estimates; (iii) the relationship between $\delta^2H_{FI}$ and $\delta^{18}O_{FI}$ may have changed over time; and (iv) the $\delta^{18}O_{FI}$ water isotope method assumes that speleothem calcite was deposited under isotopic equilibrium conditions, where isotopic non-equilibrium can either be attributed to recrystallization or kinetic isotope fractionation during calcite precipitation. The $\delta^2H_{FI}$ values are therefore the most suitable temperature proxy as they witness surface air temperature avoiding possible alterations during carbonate precipitation (Affolter et al., 2019; Uemura et al., 2020; Demény et al., 2021).

line 207: the method description is too weak. At least the TC/EA technique and the instrument should be mentioned.

In this case we have followed the methodological description commonly applied in other previous works that have used the same laboratories (Wilcox et al.,2020; Honiat et al., 2023). Even so, some extra details have been added in terms of the crushing and the TC/EA technique. This will be accompanied by the reference to the work of Dublyansky and Spötl, (2009), which contains all the specifications of the methodology used for the elaboration of this manuscript.

Line 214: Speleothem fluid inclusion water isotopes were analyzed at the University of Innsbruck using continuous-flow analysis of water via high-temperature reduction on glassy carbon in a thermal combustion/elemental analyzer (TC/EA) unit. The $\delta^2H_{FI}$ measurements were performed using a Delta V Advantage isotope ratio mass spectrometer. For details about the method of fluid inclusion preparation and isotopic analysis see Dublyansky and Spötl, (2009).

line 284: these age periods are mentioned first and last time here.

We agree with the reviewer that these terms for the different periods of the Holocene are not used again in the text. These names are now used in the new version of the manuscript to refer to the different periods of the Holocene in their discussion section.

line 292: precipitation temperature is one of the most important factors of $\delta^{18}O$ values of carbonates.

We agree that the beginning of this paragraph may imply that the main factor in the variation of $\delta^{18}O$ would be subject to the source effect. Although this is an important factor, the role of precipitation and temperature are one of the most important factors on the variability of $\delta^{18}O$ in speleothem carbonate. That is why this explanation is added to the beginning of the paragraph to clarify these important agents of change in the $\delta^{18}O$ values.

Line 303: At isotopic equilibrium, the $\delta^{18}O_c$ value is related to the $\delta^{18}O$ of the drip water and the cave temperature through its control on the equilibrium isotope fractionation between water and calcite (Lachniet, 2009). Additionally, variations in stalagmite $\delta^{18}O_c$ records may reflect changes in the $\delta18O$ of surface ocean waters from the moisture source area as well as changes in atmospheric processes which control the fractionation of oxygen isotopes in route to the site where rainfall occurs (McDermott, 2004; Lachniet, 2009).

line 301: plotting the record of seawater oxygen isotope composition would be informative.

To make it easier to avoid having to refer to other articles, we added the NISA δ18O speleothem composite, seawater oxygen isotope and sea surface salinity graphs to Fig. 6 (Fig. 6F) with their respective explanations and references.

[Figure]

line 306: there are too many pieces of information in earlier publications. Rainfall effect is mentioned here without detailed description why the given location is affected.

We agree with what the reviewer mentions here regarding the lack of an explanation of what is the main agent that controls the variation of $\delta^{18}O$ in the Mendukilo stalagmites and why this interpretation has been reached. That is why a brief reference was added at the beginning of the paragraph to how it was determined, through monitoring work in the cave, that the rainfall amount is the main agent of change in the isotopic variation of these stalagmites with a reference to the article by Bernal-Wormull et al. (2023).

Line 319: Monitoring of the Mendukilo cave suggests that the MEN δ18Oc record captured an annual signal, which is primarily influenced by rainfall amount (Bernal-Wormull et al., 2023).

lines 317-346: I miss the $\delta^2H$/T gradient numbers. It is not clear how the $\delta^2H$/T relationship was obtained. At line 337 the $\delta^2H$/T gradient seems to obtained from a two-point linear regression using MAAT ad $\delta^2H$ of drip water at two sites. On the other hand, at line 370, it is written that the $\delta^2H$/T gradient is obtained by adjusting the $\delta^{18}O$/T gradient with a factor of 8. Somehow the entire description of gradient calculation is confusing. This should be carefully revised in order to make the process more clear.

We agree with the reviewer that the derivation of a transfer function to calculate past temperatures in this work is difficult to understand and involves more steps than necessary. To correct this, we incorporate the $\delta^2H$ rainfall data and compared it to the temperature data for each event in the Fig. 2 and also we will modify the explanation of how the paleotemperature calculations were carried out. This will make possible to calculate paleotemperatures directly and it will make it easier for the reader to understand.

[Figure]

line 364: it would be easier to follow if the term δ²Hd was defined here as the δ²H value of drip water.

δ²H$_d$ is already defined as the δ²H value of drip water in the line 357.

I suggest to mention here that the equation's form expresses the fact that the pre-Holocene temperatures are lower than today's, hence the calculated temperature difference should be subtracted from T(modern).

We really appreciate Referee #1 suggestion to improve the explanation of this equation in the manuscript. We now use a sentence to mention the difference with T$_{modern}$.

Line 391: Equation (1) expresses the fact that the paleotemperatures were lower than present-day temperatures, hence the calculated temperature difference should be subtracted from T$_{modern}$.

line 372: line 105 may suggest that the δ²H/T gradient was obtained using direct monitoring data. Since both the δ¹⁸O and δ²H values monitored along with surface temperature change, this relationship can be directly calculated and it is not necessary to multiply the δ¹⁸O/T gradient by a factor of 8.

We agree with the reviewer that the calculation by a factor of 8 is not necessary to obtain the transfer function. The δ²H values of rainfall relative to MAAT will be included in Figure 2 and will be used to construct the transfer function for the calculation of paleotemperatures throughout the manuscript.

Line 393: The temperature reconstruction with Equation (1) is based on the mean relationship of 4.0‰/°C (δ²H/T$_{gradient}$; Fig. 2C) for both caves. In this case, we assessed temperature variations using speleothems from two distinct caves. As a result, it is crucial to consider the values of δ²H$_d$ and T$_{modern}$ (see chapter 2) that correspond to the δ²H$_{Fl}$ values of the speleothems from the same cave when using Equation (1). The final calculated uncertainty in the paleotemperature ranges from 1.7 to 2.6 °C (Table 3A).

line 388: the negative excursion is shown by a single point; it might derive from stochastic scatter. It can be safely written that GS-2.1a had generally cooler than GS-1.

We thank the reviewer for his recommendation on this point. The mentioned change can also be seen in the carbonate δ¹⁸O data from the Ostolo stalagmites, but it is true that the paleotemperature values to characterize this event do not support this interpretation on their own. To further support the change seen during HE1 in OM-FIT, the following points will be made clear throughout this chapter of the discussion:

1.- We provide information on why there is only one point for the calculation of paleotemperatures during HE1.

Line 415: This temperature negative excursion is supported by a single point in the δ²H$_{Fl}$ record from OST2 (Figs. 4 and 5). The same δ²H$_{Fl}$ values were obtained in further analyses of the same stalagmite and OST1, but unfortunately with water content values < 0.1 μL.

2.- The negative excursion in OST2 for this point is replicated not only in its δ²H$_{Fl}$ values, but also in what is seen in the δ¹⁸O$_c$ values of the stalagmites OST1 and OST2.

Line 418: Nevertheless, this anomaly coincides with the most negative δ¹⁸O$_c$ values in stalagmites OST1 and OST2 (Figs. 4 and 5), which corroborate that during this short period

of time in GS-2.1a there was a common catalyst in the variability of the isotopic values recorded in the speleothems of Ostolo Cave (Bernal-Wormull et al., 2021).

3.- This point is included in the conclusions, making clear the importance of future paleotemperature records of the region being able to detect this kind of centennial changes throughout the last deglaciation.

Line 632: Future quantitative temperature studies using well-dated speleothems from SW Europe are necessary to corroborate the chronology and better delineate the drastic centennial-scale temperature changes during the last deglaciation where the OM-FIT only has a few data points (e.g., HE1 and GI-1d).

Dublyansky, Y.V., and Spötl, C., 2009, Hydrogen and oxygen isotopes of water from inclusions in minerals: design of a new crushing system and on-line continuous-flow isotope ratio mass spectrometric analysis: Rapid Communications in Mass Spectrometry, v. 23, p. 2605–2613, doi:10.1002/rcm.4155.

Honiat, C., Koltai, G., Dublyansky, Y., Edwards, R.L., Zhang, H., Cheng, H., and Spötl, C., 2023, A paleoprecipitation and paleotemperature reconstruction of the Last Interglacial in the southeastern Alps: Climate of the Past, v. 19, p. 1177–1199, doi:10.5194/cp-19-1177-2023.

Wilcox, P.S., Honiat, C., Trüssel, M., Edwards, R.L., and Spötl, C., 2020, Exceptional warmth and climate instability occurred in the European Alps during the Last Interglacial period: Communications Earth & Environment, v. 1, p. 57, doi:10.1038/s43247-020-00063-w.

**Referee #2 comments (14.02.2025)**

This manuscript provides a rather interesting fluid inclusion isotope record of two caves in the western Pyrenees, that cover the time interval from Heinrich-1 to today. The isotope record is converted to a Paleo-T record, based on the present-day relation of rainfall oxygen isotope data with temperature. The resulting temperatures show a pattern of lowest T in H-1, higher T's in GI-1, lower T's again in the Younger Dryas (GS-1), and near modern temperatures for the Holocene. This is an interesting high-resolution record, providing valuable information for the reconstruction of paleotemperatures through the deglaciation in Northern Spain

I still have some comments and suggestions, however, that may improve this ms.

Dear editor,

We are grateful for the constructive comments and helpful suggestions by the reviewer. Below is a point-by-point response to the minor comments and questions by **referee #2:**

- FI isotope analysis:

The technique used for the FI isotope analysis is well-established, and an extensive dataset has been produced for this ms. This has been a formidable achievement, and provides valuable insights. The analytical uncertainty as presented, is based on the replicates, and results in a ~ 2.7 permille precision at 1SD level. Routinely analysed replicates of a standard also suggest the uncertainty is around 2.7 permille (I assume that is a 1SD statistic, but please add that to line 210).

We made an error here, and we thank the reviewer for their questions on this point.

The average long-term precision (1σ uncertainty) of replicate measurements of our in-house calcite standard is ±1.5 ‰ (and not ±2.7 ‰ as stated in the previous version) for $\delta^2H_{FI}$ for water amounts between 0.1 and 1 μL. This leads to changes in the errors involved in the calculation and the resulting paleotemperatures calculated in this work. Corresponding changes have been made to the calculations, figures, and tables with data in the appendix.

A description is added to line 220 implying that this is the standard deviation of the results.

Line 220: The average long-term precision (1σ uncertainty) of replicate measurements of an in-house calcite standard is ±2.7 ‰ for $\delta^2H_{FI}$ for water amounts between 0.1 and 1 μL.

I believe it would improve the ms to further report the d18Ofi data that were presumably collected alongside the d2Hfi data. I understand and accept that the technique used does not yield top results for d18Ofi (particularly at lower yields), but since you probably have the data, I believe it is better to show that, than to say that. Further, if part of the d18O data are robust, then d2Hfi vs d18Ofi cross-plots can help check the robustness of the analysis as a whole, and help identify analytical artefacts, like those described by Fernandez et al. (2023). I would not ask to do that in the main text, but availability of such data (and plots?) in the supplementary materials would really bolster the quality of your dataset. My stance here would be that one should not write off the d18Ofi data without trying. Quite a few

previous publications show that d18Ofi data often preserve rather well in speleothem fluid inclusions, and yield meaningful information.

In relation to that, I would strongly suggest to add a section on (isotope equilibrium) temperature calculations based on d18O calcite - d18Ofi pairs. This is a paleothermometer, used in several previous studies (e.g. Fernandez et al 2023) and has a different (isotope equilibrium) approach to temperature calculation than the one you use in this ms. Even if the d18Ofi data are not good enough, you can recalculate the d2Hfi data to d18Ofi data using the global or local MWL. I expect it may give valuable insights to have two different techniques to calculate paleotemperatures from the same set of data.

We agree with the reviewer that oxygen values from fluid inclusions may be relevant in many cases for interpretation in terms of paleotemperatures and that if they are available, they should also be included in a manuscript such as the one presented here. Even so, in this manuscript only $\delta^2H_{FI}$ values were used for calculating paleotemperatures for the following reasons: The post-depositional processes can alter the original $\delta^{18}O_{FI}$ in fluid inclusion water and thus limit the use of them for paleotemperature calculations. On the other hand, $\delta^2H_{FI}$ is not affected by isotopic fractionation during calcite precipitation and remains unaltered as there is no hydrogen source once the water is entrapped in the calcite matrix. With this setback, the decision was not to carry out the analysis of the $\delta^{18}O_{FI}$ and follow the usually methodology of other scientific articles about paleotemperatures using speleothem fluid inclusions analyzing only the $\delta^2H_{FI}$ values (Dublyansky and Spötl, 2009; Wilcox et al., 2020 Honiat et al., 2023).

- Paleothermometer calibration:

The description on how you calibrated the paleothermometer (with a modern rainfall record further E and higher up in the Pyrenees) is not so clear to me. You use a dataset of individual rain shower datapoints to make a d18O vs T relationship, and then calculate that back to d2H vs T relationship using "the factor eight" which I presume is the slope of the global MWL (or are you using a local MWL?). Several other corrections are applied (ice volume, elevation), which is correct I believe, but makes the entire calculation process a bit difficult to follow for the non-expert reader. Even when there is more info in Giménez et al (2021), I would ask to clarify this in more detail in the text of this paper, because the calculation is essential to your interpretation of the data.

Another specific question is: You calculate via d18O data now (Fig 2B), but why did you not use the d2H data of the rainwater samples, from Giménez et al 2021, straight away (because that saves you an unnecessary conversion from d18O to d2H values)?

We agree with the reviewer that the derivation of a transfer function to calculate past temperatures in this work is difficult to understand and involves more steps than necessary. To correct this, we have added the $\delta^2H$ rainfall data compared to the temperature values for each event in Fig. 2 (Fig. 2C) and also modified the explanation of how the paleotemperature calculations were carried out. This makes possible to calculate paleotemperatures directly and easier for the reader to understand.

[Figure]

- Propagation of uncertainties:

The authors state the uncertainties of the analyses and calibration are fully propagated but I could find no details on how that is done. If I look at the data underlying the d18O vs T relation (Fig 2B), then I see a significant, but seemingly not so strong correlation. The plot suggests a lot of d18O variability is not controlled by T, and I guess that should be very similar if you use the d2H data. You probably need to report the $r^2$ for the Villanúa d2H vs T plot to quantify the eventual effect on the uncertainties of the calculated T's. What I don't quite understand from the text, in that context, is how you calculated the rather good (0.03 permille) uncertainty of mean annual d18Or and MAAT.

It would be good to provide more error propagation details to underpin the uncertainties on the eventual T-data that you produce, and the choices you made to get there.

We thank reviewer 2 for identifying the lack of clarity and transparency in the calculation of errors associated with the calculation of paleotemperatures in this work. The issue of error propagation in the calculation of paleotemperatures is a difficult one to follow when considering several variables. Not only in the particular case of this manuscript, but most articles on fluid inclusions in stalagmites do not explain in depth what this concept of "propagation" of error refers to. A brief explanation of how error propagation is calculated for different types of operations that are applied in Equation (1) of the manuscript is now incorporated in the text of the Table 3A of the Appendix.

On the other hand, to obtain the value of the $\delta^{18}O/T$ it is necessary to develop a linear regression between both variables (Fig. 2). These parameters, having been obtained from experimental results, must also have an associated uncertainty. This standard error of the regression can be evaluated from the deviations between the experimental points and the predictions of the straight line. In the case of $\delta^{18}O/T$ the standard error of the regression is

0.05 (we realized that in the previous version we had made a mistake resulting in 0.03). Thanks to the comments of referee 2 we have realized that it is more direct to use the values of $\delta^2 H_r$ of above Las Güixas cave. In the case of the $\delta^2 H/T$ relationship the standard error of the regression is 0.31 and the $r^2 = 4.0$.

Thanks to the changes in the values of the transfer function, the paleotemperature values determined in this work also vary. These changes are reflected in the figures and in Tables A3 and A4.

Fernandez et al., 2023: Characterization and Correction of Evaporative Artifacts in Speleothem Fluid Inclusion Isotope Analyses as Applied to a Stalagmite from Borneo ($G^3$)

More specific comments and suggestions

286: Is that really significant? I do agree it is at the "right" point in time, but as a result this does not really stand out for me.

We believe that this result does stand out in the OM-FIT record with values very similar to those seen in the GS-1 for both caves. It should be noted that the temperature variations are not as clear as those seen in the Ostolo part of the record, but the change during the 8.2 kyr BP event is still notable, with a decrease in $\delta^2 H$ values of up to 10 ‰ compared to the average of these values during the Holocene. Furthermore, this change is visible not only in one of the stalagmites in this work, but in two of them (MEN-5 and MEN-2) with high chronological precision.

324-329: I don't quite understand what you have done here. This may need further explanation.

This paragraph refers to a statistical calculation made in the article by Gimenez et al. (2021). We agree that this is not clear in the sentence in question and seems to be a calculation made during the preparation of this manuscript, and that it is also confusing when we explain it. The sentence is corrected by giving as an example the multiple regression model of the different climatic parameters with the $\delta^2 H$ results of the rainfall, which is the isotopic indicator used to prepare the temperature transfer function to the detriment of the $\delta^{18} O$ values.

Line 342: The observed correlation between $\delta^{18} O_r$, $\delta^2 H_r$ and air temperature is verified at biannual scale above Las Güixas cave, with significant correlation between MAAT and the weighted average of $\delta^{18} O_r$ and $\delta^2 H_r$, based on a multiple regression model using a univariate Spearman's correlation of different climatic parameters (Giménez et al., 2021). For example, the correlation between $\delta^2 H_r$ and air temperature at the time of precipitation (same data series) results in a $r_s = 0.69$ (p << 0.01) by applying this multiple regression model (Giménez et al., 2021).

351: you mention Younger Dryas here, but that is not used in your figures (you use GS-1 there)

We now use only the term GS-1 along the text and the figures.

453-454 I don't immediately understand what a more smoothed T signal is? Less amplitude?

It is true that this sentence is not entirely clear. It is now clarified that it refers to the amplitude of change in isotopic values and specified which records are being compared.

Line 511: In contrast, the $\delta^{18}O_c$ of speleothems from Pyrenean caves is predominantly controlled by temperature (Bartolomé et al., 2015; Cheng et al., 2020; Bernal-Wormull et al., 2021), resulting in a subtle temperature signal (less amplitude of isotopic change) in the case of MEN $\delta^{18}O_c$ compared to the OST $\delta^{18}O_c$ record during GS-1, a cold and dry period (Fletcher et al., 2010).

477 the word "record" is once too many in this sentence.

We agree. The word "record" was erased twice in this same sentence.

482-483: masking? How does that work? Is that centennial-scale d2H variability that is not T-related?

We believe that the use of the word "masking" in this case is inappropriate and we remove it. In this paragraph our intention is to make it clear that the slight tendency towards lower temperatures during the middle Holocene cannot be discussed with the high variability of the centennial-scale $\delta^{2}H$ record and the associated errors in determining paleotemperatures.

Line 515: That 2.7 degrees C is awfully close to your T uncertainty. I would certainly not dare to say that that is more pronounced than elsewhere (like you do in line 522) based on the uncertainties you are dealing with.

We fully agree with the reviewer on this occasion. The high uncertainty in the OM-FIT does not allow for a comparison of temperature changes during the "8.2 kyr event". The sentence was modified to make this point clearer.

Line 577: The 8.2-kyr event overlapped a multi-centennial cool period from 8.29 to 8.10 ± 0.04 kyr BP recorded by MEN $\delta^{18}O_c$, characterized by an abrupt drop in temperature of about ~3.0 °C between 8.31 ± 0.06 and 8.29 ± 0.07 kyr BP in the OM-FIT record (Fig. 6). This cooling within an interglacial coincided with significant vegetation changes in the Iberian Peninsula (Allen et al., 1996; Carrión and Van Geel, 1999; González-Sampériz et al., 2006). This could be important for assessing future climate conditions in this region if changes in large parts of the climate system (climate tipping elements; Armstrong McKay et al., 2022) intensify beyond a warming threshold.

The cooling amplitude during the 8.2 kyr event recorded by OM-FIT appears at first glance more pronounced than in other Northern Hemisphere temperature and precipitation records, with proxy evidence across Europe indicating a cooling by ~ 1-1.7 °C during this event (Davis et al., 2003; Morrill et al., 2013; Baldini et al., 2019), but unfortunately the uncertainties involved in this records and the OM-FIT do not allow a valid comparison to be made on this occasion.

Table A1: reporting reproducibility with a 1SD metric on only two replicates is statistically a bit on the edge, perhaps.

We agree with the reviewer that reporting the standard deviation of a point with only two measurements is not ideal. Still, it is a common practice in this type of records (see Wilcox et al., 2020; Honiat et al., 2023) where the material is scarce and where the amounts of water to be analyzed are mostly too small to be considered reliable for determining paleotemperatures. In this case, the values are later compared with carbonate isotopes of both caves, which further supports the variability seen in the $\delta^2H$ values of the fluid inclusions throughout the last deglaciation.

Fig 4 panel B: Y axis labels are in the wrong place (swapped).

We fixed the titles of the axis in Fig. 4B.

Dublyansky, Y.V., and Spötl, C., 2009, Hydrogen and oxygen isotopes of water from inclusions in minerals: design of a new crushing system and on-line continuous-flow isotope ratio mass spectrometric analysis: Rapid Communications in Mass Spectrometry, v. 23, p. 2605–2613, doi:10.1002/rcm.4155.

Honiat, C., Koltai, G., Dublyansky, Y., Edwards, R.L., Zhang, H., Cheng, H., and Spötl, C., 2023, A paleoprecipitation and paleotemperature reconstruction of the Last Interglacial in the southeastern Alps: Climate of the Past, v. 19, p. 1177–1199, doi:10.5194/cp-19-1177-2023.

Wilcox, P.S., Honiat, C., Trüssel, M., Edwards, R.L., and Spötl, C., 2020, Exceptional warmth and climate instability occurred in the European Alps during the Last Interglacial period: Communications Earth & Environment, v. 1, p. 57, doi:10.1038/s43247-020-00063-w.